# `CurvGAD`: Leveraging Curvature for Enhanced Graph Anomaly Detection

**Karish Grover** [1]  **Geoffrey J. Gordon** [1]  **Christos Faloutsos** [1]

## Abstract

Does the intrinsic *curvature* of complex networks hold the key to unveiling graph anomalies that conventional approaches overlook? Reconstruction-based graph anomaly detection (GAD) methods overlook such *geometric* outliers, focusing only on structural and attribute-level anomalies. To this end, we propose `CurvGAD` — a mixed-curvature graph autoencoder that introduces the notion of curvature-based geometric anomalies. `CurvGAD` introduces two parallel pipelines for enhanced anomaly interpretability: (1) *Curvature-equivariant geometry reconstruction*, which focuses exclusively on reconstructing the edge curvatures using a mixed-curvature, Riemannian encoder and Gaussian kernel-based decoder; and (2) *Curvature-invariant structure and attribute reconstruction*, which decouples structural and attribute anomalies from geometric irregularities by regularizing graph curvature under discrete Ollivier-Ricci flow, thereby isolating the non-geometric anomalies. By leveraging curvature, `CurvGAD` refines the existing anomaly classifications and identifies new curvature-driven anomalies. Extensive experimentation over 10 real-world datasets (both homophilic and heterophilic) demonstrates an improvement of up to **6.5%** over state-of-the-art GAD methods. The code is available at: https://github.com/karish-grover/curvgad.

## 1. Introduction

Detecting anomalies in graph-structured data is a pivotal task across diverse fields, including social networks (Hassanzadeh et al., 2012), cybersecurity (Wang & Zhu, 2022), transportation systems (Hu et al., 2020), and biological networks (Singh & Vig, 2017). Traditional methods primarily focus on identifying anomalies through structural irregularities or attribute deviations (Ding et al., 2019; Fan et al., 2020), such as abnormal connections or unusual feature values. However, these approaches often overlook the underlying geometric properties of graphs, particularly the graph *curvature*, which encapsulates essential information about the global and local topology of the graph. Recent advances in graph representation learning and Riemannian geometry have delved deeply into constant *curvature* spaces (Bachmann et al., 2020) — namely hyperbolic, spherical, and Euclidean geometries, characterized by negative, positive, and zero curvature, respectively — to learn distortion-free graph representations. Each of these spaces offers beneficial inductive biases for specific structures. For instance, hyperbolic space is ideal for hierarchical, tree-like graphs (Chami et al., 2019), whereas spherical geometry is optimal for representing cyclic graphs (Gu et al., 2019). However, current graph anomaly detection (GAD) methods fail to leverage such insights, resulting in several limitations:

(L1) **Inadequate Representation of Complex Topologies**. Existing methods assume that graphs can be effectively represented in Euclidean space. However, real-world graphs often exhibit complex topologies with intrinsic curvature variations (Gu et al., 2019) that cannot be captured by a single geometric space.

(L2) **Neglect of Geometric Anomalies**. Current approaches largely ignore *geometric anomalies* manifested through task-specific irregularities in the curvature of the graph, missing critical insights into the inherent geometry of the graph (Chatterjee et al., 2021).

(L3) **Homophily Trap**. Several methods predominantly operate under the homophily assumption (i.e., low-pass filters) and fail in heterophilic graphs where connected nodes tend to be dissimilar, missing out on anomalies that arise in such settings (He et al., 2023).

*Geometric anomalies*, as we define them, are irregularities in graph structure revealed through deviations in task-specific curvature patterns. Depending on the application, different curvature regimes — negative, positive, or zero curvature — may indicate anomalies. These anomalies occur when the curvature at certain nodes or edges significantly deviates from expected patterns, signaling unusual structural properties or interactions not detectable through traditional methods. There are numerous real-world instances where

[1]Carnegie Mellon University. Correspondence to: Karish Grover <karishg@cs.cmu.edu>.

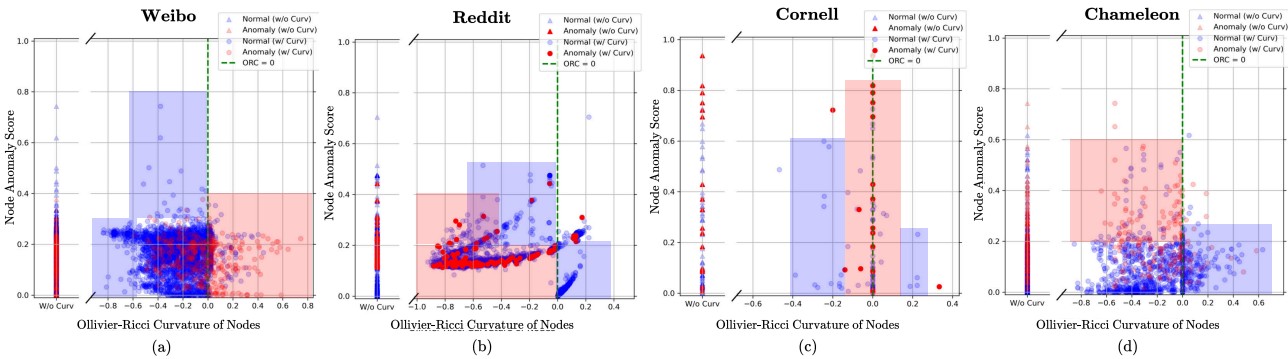

*Figure 1.* **Curvature matters**. The effect of curvature information on node-level anomaly scores for `CurvGAD` across Weibo, Reddit, Cornell and Chameleon datasets (Zhao et al., 2020). The plots show anomaly score distributions against curvature values, where the blue region highlights potentially new anomalies due to curvature deviations, and the red region highlights the effect of curvature on existing outliers. *W/o Curv* refers to anomaly scores calculated without curvature. Observe that interesting anomaly-curvature patterns arise – e.g., in (a), *known* anomalies exhibit predominantly positive curvature, whereas in (d), they are mostly negative. In (b), some non-anomalous nodes exhibit irregular curvature deviations, forming an isolated cluster (i.e. potential curvature anomalies).

curvature can potentially serve as a critical heuristic for detecting anomalies in graph-structured data, e.g. **(a)** *Fake News Propagation*: Hierarchical cascade-like diffusion patterns in social networks, often the result of fake news dissemination by malicious users (anomalies), exhibit high negative curvature due to rapid branching and sparse connections (Ducci et al., 2020; Xu et al., 2021). **(b)** *Biological Networks*: Bottleneck proteins in protein-protein interaction (PPI) networks, acting as critical hubs between functional modules, signal disruptions that can affect biological processes, and have positive curvature (Topping et al., 2021).

We propose `CurvGAD`, a novel framework for detecting graph anomalies by integrating geometric insights through a mixed-curvature lens. Intuitively, `CurvGAD` addresses the limitations of current GAD methods in two key ways: **(1)** It refines the classification of pre-existing anomalies by incorporating curvature information, improving the detection of structural or attribute-level anomalies by leveraging geometric insights; and **(2)** It uncovers previously undetected anomalies driven by curvature irregularities — anomalies that may not be labeled in the original dataset but emerge through curvature deviations. The core idea behind `CurvGAD` is the decomposition of graph anomalies into two parallel pipelines described below.

**(a) Curvature-equivariant Geometry Reconstruction**. Detects geometric anomalies by learning representations in mixed-curvature spaces and reconstructing the graph's curvature matrix (addresses limitation L2). The encoder employs a mixed-curvature Chebyshev polynomials-based filter bank, which encodes graph signals into representations that adapt to the curvature of the underlying graph topology (addresses L1). This includes multiple low-pass and high-pass filters operating in a product manifold, ensuring that we capture signals from multiple bands in the eigen-spectrum (addresses L3). Using the node embeddings, the decoder applies a Gaussian kernel-based approximation to reconstruct the edge curvatures. Geometric anomalies manifest as irregular curvature values, typically associated with a relatively larger reconstruction loss, providing insights into anomalies such as bottlenecks and hubs within the graph.

**(b) Curvature-invariant Structure and Attribute Reconstruction**. Reconstructs the adjacency and feature matrices, ensuring that the process remains invariant to the curvature of the graph. To achieve this, the input graph is first regularized by deforming it under the discrete Ollivier-Ricci flow, which standardizes the curvature of the graph, converging to a uniform value. This allows the subsequent structure and attribute reconstructions to focus solely on non-geometric anomalies. The encoder operates on the regularized graph, while the decoder reconstructs the adjacency and feature matrices. Decoupling geometric irregularities from the non-geometric ones, ensures a dedicated focus on the former (addresses L2). This unified framework improves the ability to detect a broader spectrum of graph anomalies. Our key contributions can be summarized as follows.

(C1) **Novelty:** To the best of our knowledge, this is the **first** work to study curvature-based anomalies and approach GAD from a mixed-curvature perspective.

(C2) **Interpretability:** Offers interpretable detection by disentangling curvature-induced anomalies from structural and attribute-level irregularities.

(C3) **Universality:** `CurvGAD` performs well in detecting geometric, structural, and attribute-based anomalies across heterophilic and homophilic networks.

(C4) **Effectiveness:** Experimentation with 10 real-world datasets for node-level GAD shows that `CurvGAD` achieves up to **6.5% gain** over SOTA methods.

## 2. Previous Works

**Graph Anomaly Detection**. GAD aims to identify abnormal patterns or instances in graph data. Traditional node-level GAD methods focus primarily on detecting (a) *structural* or (b) *attribute-based* anomalies within a graph. Recent advances have introduced reconstruction-based approaches for GAD. These methods employ autoencoders to reconstruct graph structures (adjacency matrices) and node attributes, characterizing anomalies with a higher reconstruction error as anomalous nodes or substructures are relatively difficult to reconstruct. For example, DOMINANT (Ding et al., 2019) and AnomalyDAE (Fan et al., 2020) use a dual discriminative mechanism to simultaneously detect structural and attribute anomalies by minimizing reconstruction loss in the adjacency and feature matrices.

**Riemannian Graph Neural Networks**. Non-Euclidean manifolds, particularly hyperbolic (Sala et al., 2018) and spherical (Liu et al., 2017) geometries, have proven effective for learning distortion-minimal graph representations. Two main approaches dominate this domain: (a) *Single Manifold GNNs*: Models like HGAT (Zhang et al., 2021) and HGCN (Chami et al., 2019) achieve state-of-the-art performance on hierarchical graphs by embedding them in hyperbolic spaces. (b) *Mixed-Curvature GNNs*: Recognizing that single-geometry manifolds fall short in representing complex, real-world topologies, mixed-curvature GNNs embed graphs in product manifolds combining spherical, hyperbolic, and Euclidean components. Pioneered by (Gu et al., 2019), this idea has been extended by models like $\kappa$-GCN (Bachmann et al., 2020), which uses the $\kappa$-stereographic model, and Q-GCN (Xiong et al., 2022), which operates on pseudo-Riemannian manifolds. Despite these efforts, GAD methods overlook geometric information (curvature), spectral properties (e.g., heterophily), and fail to leverage Riemannian embeddings, limiting their ability to detect more nuanced anomalies that arise in complex graph structures.

## 3. Preliminaries

**Riemannian Geometry**. A smooth *manifold* $\mathcal{M}$ generalizes surfaces to higher dimensions. At each point $\mathbf{x} \in \mathcal{M}$, the *tangent space* $\mathcal{T}_{\mathbf{x}}\mathcal{M}$ is locally Euclidean. The *Riemannian metric* $g_{\mathbf{x}}(\cdot,\cdot) : \mathcal{T}_{\mathbf{x}}\mathcal{M} \times \mathcal{T}_{\mathbf{x}}\mathcal{M} \to \mathbb{R}$ equips $\mathcal{T}_{\mathbf{x}}\mathcal{M}$ with an inner product that enables the definition of distances and angles, forming a *Riemannian manifold* (Do Carmo & Flaherty Francis, 1992). The *exponential map* $\exp_{\mathbf{x}}(\mathbf{v}) : \mathcal{T}_{\mathbf{x}}\mathcal{M} \to \mathcal{M}$ maps tangent vectors to the manifold, while the *logarithmic map* $\log_{\mathbf{x}}(\mathbf{y}) : \mathcal{M} \to \mathcal{T}_{\mathbf{x}}\mathcal{M}$ maps points back to the tangent space. The curvature ($\kappa$) at each point describes the geometry, with three common types: positively curved *spherical* ($\mathbb{S}$) ($\kappa > 0$), negatively curved *hyperbolic* ($\mathbb{H}$) ($\kappa < 0$), and flat *Euclidean* ($\mathbb{E}$) ($\kappa = 0$).

**Product Manifolds**. A product manifold (Gu et al., 2019) $\mathbb{P}$ is defined as the Cartesian product of $\mathcal{P}$ constant-curvature manifolds, i.e. $\mathbb{P} = \times_{p=1}^{\mathcal{P}} \mathcal{M}_p^{\kappa_p, d_p}$, where $\mathcal{M}_p \in \{\mathbb{E}, \mathbb{H}, \mathbb{S}\}$ represents a component manifold with curvature $\kappa_p$ and dimension $d_p$. The total dimension of the product manifold is the sum of the component dimensions. The above decomposition of $\mathbb{P}$ is called its *signature* (Appendix B.1).

$\kappa-$**Stereographic Model**. In this work, we adopt the $\kappa$-Stereographic Model (Bachmann et al., 2020) to define Riemannian algebraic operations across both positively and negatively curved spaces within a unified framework. This model eliminates the need for separate mathematical formulations for different geometries. In particular, $\mathcal{M}^{\kappa,d}$ is the stereographic sphere model for spherical manifold ($\kappa > 0$), while it is the Poincaré ball model (Ungar, 2001) for hyperbolic manifold ($\kappa < 0$) (See Appendix B.2).

**Discrete Laplace-Beltrami (LB) Operator**. The Laplace–Beltrami operator (Urakawa, 1993) is a generalization of the Laplace operator to functions defined on Riemannian manifolds. We define the discrete Laplace-Beltrami operator, $\mathbf{L}_{\mathbb{P}}$, for a graph discretized over the product manifold $\mathbb{P}$, based on the *cotangent* discretization scheme (Belkin et al., 2008; Crane, 2019). For two connected vertices $v_i$ and $v_j$, the off-diagonal element of $\mathbf{L}_{\mathbb{P}}$ is: $\mathbf{L}_{\mathbb{P},ij} = -\frac{(\cot \theta_{ij} + \cot \phi_{ij})}{2A_i}$ where $\theta_{ij}$ and $\phi_{ij}$ represent the angles opposite to the edge $(i, j)$ in the adjacent triangles, and $A_i$ is the Voronoi area of vertex $v_i$ (or Heron's area in the case of obtuse triangles). The diagonal element is computed as: $\mathbf{L}_{\mathbb{P},ii} = -\sum_{j \neq i} \mathbf{L}_{\mathbb{P},ij}$ This operator highlights the geometric properties by incorporating the manifold curvature.

**Ollivier-Ricci Curvature**. In graphs, the lack of an inherent manifold structure necessitates the use of discrete curvature analogs, such as Ollivier-Ricci curvature (ORC) (Ollivier, 2007), which extends the concept of continous manifold curvature (Tanno, 1988) to networks. ORC is defined as a transportation-based curvature along an edge, where the curvature between neighborhoods of two nodes is measured via the Wasserstein-1 distance (Piccoli & Rossi, 2016). For an unweighted graph, each node $x$ is assigned a neighborhood measure $m_x^\delta(z) := \frac{1-\delta}{|\mathcal{N}(x)|}$ for $z \in \mathcal{N}(x)$, and $m_x^\delta(x) = \delta$. ORC for an edge $(x, y)$ is then calculated as: $\widetilde{\kappa}_{xy} := 1 - \frac{\mathbf{W}_1(m_x^\delta, m_y^\delta)}{d_{\mathcal{G}}(x,y)}$. We approximate ORC in linear time using combinitorial bounds (Jost & Liu, 2014). See Appendix B.3 for details on computational considerations. We denote the continuous manifold curvature using $\kappa$ and the discrete Ollivier-Ricci curvature using $\widetilde{\kappa}$.

**Ollivier-Ricci Flow**. The Ricci flow, introduced by Hamilton (Chow et al., 2023), is a process that smooths the curvature of a manifold by deforming its metric over time. In the graph domain, this concept is adapted to the discrete Ollivier-Ricci flow. In each iteration of this evolving process,

Ricci flow generates a time-dependent family of weighted graphs $(\mathcal{V}, \mathcal{E}, \mathbf{w}^{(t)})$ such that the weight $\mathbf{w}_{xy}^{(t)}$ on edge $xy$ changes proportionally to the ORC of the edge $xy$ at time $t$, $\widetilde{\kappa}_{xy}^{(t)}$. Ollivier (Ollivier, 2009) defined the Ricci flow for continuous time as $\frac{d}{dt}\mathbf{w}_{xy}^{(t)} = -\widetilde{\kappa}_{xy}^{(t)} \cdot \mathbf{w}_{xy}^{(t)}$. Then (Ni et al., 2018) proposed Ricci flow for discrete time $t$ (for step size $\epsilon$) as: $\mathbf{w}_{xy}^{(t+1)} = (1 - \epsilon \widetilde{\kappa}_{xy}^{(t)})\mathbf{w}_{xy}^{(t)}$. For an unweighted graph, $\mathbf{w}_{xy}^{(0)} = 1, \; \forall x, y \in \mathcal{V}$. Typically, ORC for all edges converges to zero curvature as $t \to \infty$.

**Setup.** We consider graphs $\mathcal{G} = (\mathcal{V}, \mathcal{E}, \mathbf{A})$, where $\mathcal{V}$ is a set of $|\mathcal{V}| = n$ vertices, $\mathcal{E}$ is a set of edges, and $\mathbf{A} \in \mathbb{R}^{n \times n}$ is the adjacency matrix. The nodes are associated with the node feature matrix $\mathbf{X} \in \mathbb{R}^{n \times d_\mathcal{X}}$ ($d_\mathcal{X}$ is the feature dimension). A graph signal $\mathbf{x} : \mathcal{V} \to \mathbb{R}$ may be regarded as a vector $\mathbf{x} \in \mathbb{R}^n$ where $x_i$ is the value of $\mathbf{x}$ at the $i^{th}$ node. The Laplace-Beltrami operator $\mathbf{L}_\mathbb{P}$ (Section 3) extends the notion of the traditional graph Laplacian to mixed-curvature spaces by incorporating geometric clues. As a real, symmetric, and positive semidefinite matrix, $\mathbf{L}_\mathbb{P}$ admits a complete set of orthonormal eigenvectors $\mathbf{U}_\mathbb{P} = [\{\mathbf{u}_l\}_{l=0}^{n-1}] \in \mathbb{R}^{n \times n}$, and their associated ordered real nonnegative eigenvalues $[\{\lambda_l\}_{l=0}^{n-1}] \in \mathbb{R}^n$, identified as the *frequencies* of the graph. Similar to the graph Laplacian, the Laplace-Beltrami matrix can be diagonalized as $\mathbf{L}_\mathbb{P} = \mathbf{U}_\mathbb{P} \mathbf{\Lambda}_\mathbb{P} \mathbf{U}_\mathbb{P}^\top$ where $\mathbf{\Lambda}_\mathbb{P} = \mathbf{diag}([\{\lambda_l\}_{l=0}^{n-1}]) \in \mathbb{R}^{n \times n}$.

## 4. Proposed Approach: CurvGAD

In this section, we provide an in-depth exploration of the CurvGAD architecture. We first introduce the curvature-equivariant pipeline (Section 4.1), which reconstructs the curvature matrix to detect geometric anomalies. This involves a mixed-curvature encoder (common to both pipelines) equipped with spectral graph filters (Section 4.1.1), followed by a Gaussian kernel-based decoder (Section 4.1.2) that predicts curvature values. Next, we extend our framework to curvature-invariant anomaly detection (Section 4.2) by deforming the graph under Ollivier-Ricci flow (Section 4.2.1), thereby regularizing curvature distortions. While the encoder remains identical, we replace the product manifold with a Euclidean manifold to ensure that structural and attribute anomalies are reconstructed independently of curvature.

### 4.1. Curvature-equivariant Reconstruction

**Product manifold Construction**. The product manifold $\mathbb{P}$ can have multiple hyperbolic or spherical components with distinct *learnable* curvatures. This allows us to accommodate a broader spectrum of curvatures. Consequently, the product manifold can be succinctly described as $\mathbb{P}^{d_\mathcal{P}} = \times_{p=1}^{\mathcal{P}} \mathcal{M}_p^{\kappa_p, d_p} = (\times_{h=1}^{\mathcal{H}} \mathbb{H}_h^{\kappa_h, d_h}) \times (\times_{s=1}^{\mathcal{S}} \mathbb{S}_s^{\kappa_s, d_s}) \times \mathbb{E}^{d_e}$, with a total dimension of $d_\mathcal{P} = \sum_{h=1}^{\mathcal{H}} d_h + \sum_{s=1}^{\mathcal{S}} d_s + d_e =$

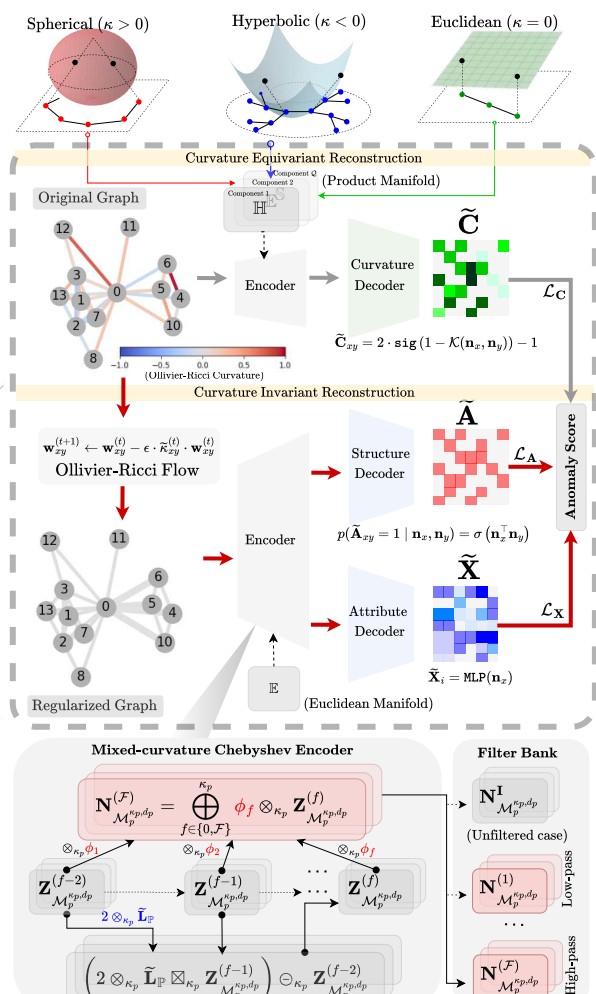

*Figure 2.* **Architecture of CurvGAD.** The proposed model employs two synergistic pipelines for anomaly detection: **(1)** *Curvature-Equivariant Reconstruction* – embedding the input graph into a mixed-curvature product manifold using a spectral Chebyshev filter bank, with a Gaussian kernel-based decoder reconstructing the curvature matrix to capture curvature irregularities; **(2)** *Curvature-Invariant Reconstruction* – regularizing the graph under Ollivier-Ricci flow to uniformize edge curvatures, followed by an Euclidean manifold-based encoder to learn representations for the decoupled reconstruction of adjacency and feature matrices.

$\sum_{p=1}^{\mathcal{P}} d_p$. We determine the task-specific signature of $\mathbb{P}^{d_\mathcal{P}}$, by examining the distribution of the Ollivier-Ricci curvature within the graph and identifying the most significant curvature bands (see Appendix F). Next, we project the original Euclidean input node features onto the mixed-curvature manifold as $\mathbf{X}' = \|_{p=1}^{\mathcal{P}} \exp_0^{\kappa_p}(f_\theta(\mathbf{X}))$, where $f_\theta(.) : \mathbb{R}^{d_\mathcal{X}} \to \mathbb{R}^{d_\mathcal{P}}$ represents a neural network with parameter set $\{\theta\}$ that generates the hidden state euclidean features of dimension $d_\mathcal{P}$. Here, $\exp_0^{\kappa_p} : \mathbb{R}^{d_\mathcal{P}} \to \mathcal{M}^{\kappa_p, d_\mathcal{P}}$ is the exponential map (Section 3) to project the node features $\mathbf{X}$ to the $p^{th}$ manifold. The projected features are

utilized in the encoder to learn node representations, which are subsequently used to reconstruct the curvature.

### 4.1.1. MIXED-CURVATURE CHEBYSHEV ENCODER

Once the input features are projected into the product manifold, we introduce a filterbank of Chebyshev approximation-based spectral filters. Let $\psi$ represent the graph filter operator, the filtering operation for a signal $\mathbf{x}$ is defined as:

$$\psi(\mathbf{L}_{\mathbb{P}})\mathbf{x} = \psi(\mathbf{U}_{\mathbb{P}}\mathbf{\Lambda}_{\mathbb{P}}\mathbf{U}_{\mathbb{P}}^{\top})\mathbf{x} = \mathbf{U}_{\mathbb{P}}\psi(\mathbf{\Lambda}_{\mathbb{P}})\mathbf{U}_{\mathbb{P}}^{\top}\mathbf{x}. \quad (1)$$

■ **Chebyshev Approximation.** Direct eigen- decomposition of the Laplace-Beltrami operator $\mathbf{L}_{\mathbb{P}}$ is computationally prohibitive, especially for large graphs. To address this we employ a Chebyshev polynomial approximation of the filters *in the mixed-curvature space*. Recall that the Chebyshev polynomial $Z^{(f)}(x)$ of order $f$ may be computed by the following stable recurrence relation:

$$Z^{(f)}(x) = 2Z^{(f-1)}(x) - Z^{(f-2)}(x), \quad (2)$$

with $Z^{(0)} = 1$ and $Z^{(1)} = x$. In our framework, we generalize this recurrence to accommodate the geometry of the mixed-curvature manifold by replacing the standard operations with their curvature-aware counterparts. The filtering process on the $p^{\text{th}}$ manifold proceeds as follows:

$$\mathbf{Z}_{\mathcal{M}_p^{\kappa_p,d_p}}^{(0)} = \exp_{\mathbf{0}}^{\kappa_p}(f_\theta(\mathbf{X})) \quad \text{(Feature mapping)} \quad (3)$$

$$\mathbf{Z}_{\mathcal{M}_p^{\kappa_p,d_p}}^{(1)} = \widetilde{\mathbf{L}}_{\mathbb{P}} \boxtimes_{\kappa_p} \exp_{\mathbf{0}}^{\kappa_p}(f_\theta(\mathbf{X})) \quad (4)$$

$$\mathbf{Z}_{\mathcal{M}_p^{\kappa_p,d_p}}^{(f)} = \left(2 \otimes_{\kappa_p} \widetilde{\mathbf{L}}_{\mathbb{P}} \boxtimes_{\kappa_p} \mathbf{Z}_{\mathcal{M}_p^{\kappa_p,d_p}}^{(f-1)}\right) \ominus_{\kappa_p} \mathbf{Z}_{\mathcal{M}_p^{\kappa_p,d_p}}^{(f-2)} \quad (5)$$

$$\mathbf{N}_{\mathcal{M}_p^{\kappa_p,d_p}}^{(\mathcal{F})} = \bigoplus_{f \in \{0,\mathcal{F}\}}^{\kappa_p} \phi_f \otimes_{\kappa_p} \mathbf{Z}_{\mathcal{M}_p^{\kappa_p,d_p}}^{(f)} \quad (6)$$

Observe how Equations 3 to 5 generalise the Chebyshev recurrence in Equation 2 to the mixed-curvature setting, for the $p^{\text{th}}$ component manifold. Further, as detailed in a previous section, the Equation 3 transforms the original Euclidean node features $\mathbf{X}$ to the product manifold. Finally, in Equation 6 the filtered signals from different Chebyshev orders (till $\mathcal{F}$) are aggregated to form the final node representation on the manifold $p$. Here, $\oplus_\kappa$, $\ominus_\kappa$, $\otimes_\kappa$ and $\boxtimes_\kappa$ denote *mobius* addition, *mobius* subtraction, $\kappa$-right-matrix-multiplication and $\kappa$-left-matrix-multiplication respectively (Appendix B.2). These operations generalize vector algebra to $\kappa$- stereographic model. $\phi_l$ are the learnable weights.

■ **Filter Bank.** To effectively capture spectral information across diverse graph frequencies, we adopt the above construction to get a filter bank of multiple spectral filters:

$$\Omega_{\mathcal{M}_p^{\kappa_p,d_p}} = \left[\mathbf{N}_{\mathcal{M}_p^{\kappa_p,d_p}}^{\mathbf{I}}, \mathbf{N}_{\mathcal{M}_p^{\kappa_p,d_p}}^{(1)}, \ldots, \mathbf{N}_{\mathcal{M}_p^{\kappa_p,d_p}}^{(\mathcal{F})}\right]$$

Here, $\mathbf{N}^{\mathbf{I}}$ is the unfiltered case, where we pass the identity matrix $\mathbf{I}$ instead of $\widetilde{\mathbf{L}}_{\mathbb{P}}$. This approach ensures that the filter bank captures both high-frequency (heterophilic) and low-frequency (homophilic) signals within the mixed-curvature manifold, enabling CurvGAD to generalize effectively across heterophilic and homophilic graphs.

■ **Final Representations.** Filtered node representations are aggregated hierarchically to synthesize information across both spectral filters and manifold components. Unlike naive concatenation, this aggregation mechanism assigns learnable importance weights to both filters and manifolds:

$$\mathbf{n}_j = \left\|_{m=0}^{\mathcal{M}} \beta_p \otimes_{\kappa_p} \left(\bigoplus_{f=0}^{\mathcal{F}} \alpha_f \otimes_{\kappa_p} \left(\mathbf{N}_{\mathcal{M}_p^{\kappa_p,d_p}}^{(f)}\right)\right)_j\right.$$

Here $\|$ is the concatenation operator and $\mathbf{n}_j \in \mathbb{P}^{d_\mathcal{P}}$ is final node representation for node $j$. $\beta_p$ and $\alpha_f$ are learnable weights which assert the relative importance of the $p^{th}$ component manifold embedding and $f^{th}$ filter.

### 4.1.2. DECODER FOR CURVATURE RECONSTRUCTION

The decoder aims to reconstruct the Ollivier-Ricci curvature $\mathbf{C}_{xy}$ for nodes $x$ and $y$, by leveraging their latent embeddings $\mathbf{n}_x, \mathbf{n}_y \in \mathbb{P}^{d_\mathcal{M}}$, learned through the mixed-curvature encoder. We propose using a *Gaussian kernel* defined on the manifold distance between the node embeddings.

**Definition 4.1** (*Curvature Decoder*). Let $\mathbf{n}_x, \mathbf{n}_y \in \mathbb{P}^{d_\mathcal{M}}$ be the latent embeddings of nodes $x$ and $y$. The predicted curvature $\widetilde{\mathbf{C}}_{xy}$ is defined as: $\widetilde{\mathbf{C}}_{xy} = 2 \cdot \texttt{sigmoid}(1 - \mathcal{K}(\mathbf{n}_x, \mathbf{n}_y)) - 1$, where $\mathcal{K}(\mathbf{n}_x, \mathbf{n}_y)$ is a Gaussian kernel: $\mathcal{K}(\mathbf{n}_x, \mathbf{n}_y) = \exp\left(-\gamma \frac{\mathcal{D}_\mathcal{M}(\mathbf{n}_x, \mathbf{n}_y)^2}{\tau^2}\right)$, and $\mathcal{D}_\mathcal{M}(\mathbf{n}_x, \mathbf{n}_y) = \sqrt{\sum_{m=1}^{\mathcal{M}} \mathcal{D}_{\mathcal{M}_p}(\mathbf{n}_x, \mathbf{n}_y)^2}$ aggregates geodesic distances $\mathcal{D}_{\mathcal{M}_p}$ over manifold components. Here, $\gamma$ is a fixed kernel width and $\tau$ is a sensitivity parameter.

See Appendix D.1 for more intuition. The decoder minimizes the loss of the Frobenius norm: $\mathcal{L}_\mathbf{C} = \|\widetilde{\mathbf{C}} - \mathbf{C}\|_F^2$, where $\widetilde{\mathbf{C}}$ and $\mathbf{C}$ denote the predicted and original curvature matrices, respectively. This objective ensures accurate curvature reconstruction while adapting to geodesic distances.

### 4.2. Curvature-invariant Reconstruction

This pipeline focuses on reconstructing the adjacency matrix $\mathbf{A}$ and the feature matrix $\mathbf{X}$ independently of the graph's underlying geometry (curvature). To achieve this, the graph is first deformed under the Ollivier-Ricci flow, followed by curvature-invariant encoding and decoding.

### 4.2.1. RICCI FLOW AND CURVATURE REGULARIZATION

Ollivier-Ricci flow is applied to the original graph to regularize edge curvatures by iteratively updating edge weights

based on their curvature values. This process, outlined in Algorithm 4.2.1, transforms the graph into a constant-curvature space, thereby neutralizing curvature-induced distortions. Refer to Figure 3 for better intuition.

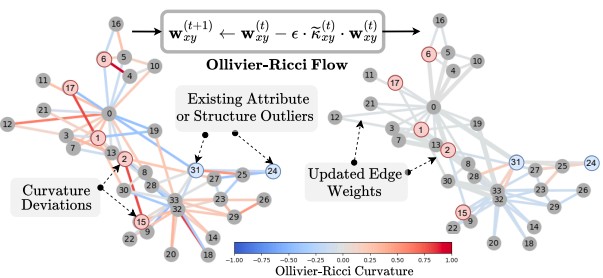

*Figure 3.* **Intuition on Ollivier-Ricci flow.** Curvature regularization of Karate Club graph (Rozemberczki et al., 2020) under Ollivier-Ricci flow. Observe how the flow increases (*decreases*) the weight of negatively (*positively*) curved edges. Red nodes indicate curvature-based outliers near edges with extreme curvatures.

---

**Algorithm 1** Discrete Ollivier-Ricci Flow

**Require:** An undirected graph $\mathcal{G} = (\mathcal{V}, \mathcal{E})$, a small threshold $\Delta > 0$, learning rate $\epsilon > 0$
**Ensure:** Edge weights of $\mathcal{G}$ as Ricci flow metrics

1: Initialize edge weights $\mathbf{w}_{xy}^{(0)} = 1$ for all $(x, y) \in \mathcal{E}$
2: **repeat**
3:      Normalize edge weights: $\mathbf{w}_{xy}^{(t)} \leftarrow \frac{|\mathcal{E}| \cdot \mathbf{w}_{xy}^{(t)}}{\sum_{(x,y)\in\mathcal{E}} \mathbf{w}_{xy}^{(t)}}$
4:      Compute ORC: $\widetilde{\kappa}_{xy}^{(t)} = 1 - \frac{\mathbf{W}_1(m_x^\delta, m_y^\delta)}{\mathcal{D}_\mathcal{G}^{(t)}(x,y)}$
5:      Update edge weight under Ricci flow:

$$\mathbf{w}_{xy}^{(t+1)} \leftarrow \mathbf{w}_{xy}^{(t)} - \epsilon \cdot \widetilde{\kappa}_{xy}^{(t)} \cdot \mathbf{w}_{xy}^{(t)}$$

6:      Check convergence: $\left| \widetilde{\kappa}_{xy}^{(t+1)} - \widetilde{\kappa}_{xy}^{(t)} \right| < \Delta$
7: **until** convergence condition is met for all edges

---

### 4.2.2. UNIFORM-CURVATURE RECONSTRUCTION

Under Ollivier-Ricci flow, the edge curvatures of the *regularized* graph approach near zero, allowing reconstructions to operate within a Euclidean manifold. We reuse the Chebyshev filterbank encoder (Section 4.1.1) to map the nodes in the curvature-regularized graph (Section 4.2.1) to latent Euclidean embeddings, ensuring curvature invariance in the learned representations. Unlike the mixed-curvature encoder (Section 4.1), this encoder simplifies the geometric complexity by restricting operations to a Euclidean space.

■ **Decoders**. We employ separate decoders for adjacency and feature reconstruction tasks. These decoders ensure that structural and attribute information is effectively recovered without being influenced by curvature irregularities.

**Definition 4.2** (*Adjacency Decoder*)**.** The predicted adjacency value $\widetilde{\mathbf{A}}_{xy}$ for nodes $x, y$ is modeled as the likelihood of an edge existing between them: $p(\mathbf{A}_{xy} = 1 \mid \mathbf{n}_x, \mathbf{n}_y) = \sigma\left(\mathbf{n}_x^\top \mathbf{n}_y\right)$, where $\mathbf{n}_x, \mathbf{n}_y \in \mathbb{R}^d$ are Euclidean latent embeddings, and $\sigma(\cdot)$ is the sigmoid activation.

**Definition 4.3** (*Attribute Decoder*)**.** The reconstructed features for a node x are given by: $\widetilde{\mathbf{X}}_x = f_{\text{dec-X}}(\mathbf{n}_x)$, where $f_{\text{dec-X}}$ is a multi-layer perceptron (MLP) mapping latent embeddings $\mathbf{n}_x \in \mathbb{R}^d$ to feature space.

The decoders are trained using reconstruction losses: $\mathcal{L}_{\mathbf{A}} = \|\widetilde{\mathbf{A}} - \mathbf{A}\|_F^2$ for adjacency and $\mathcal{L}_{\mathbf{X}} = \|\widetilde{\mathbf{X}} - \mathbf{X}\|_F^2$ for features.

### 4.3. Objective Function

The objective function balances curvature reconstruction, structural and attribute reconstruction, and a supervised classification loss to detect both geometric and non-geometric anomalies. The total loss is:

$$\mathcal{L}_{\text{total}} = \lambda_{\text{cls}}\mathcal{L}_{\text{cls}} + (1 - \lambda_{\text{cls}}) \cdot (\lambda_{\mathbf{C}}\mathcal{L}_{\mathbf{C}} + \lambda_{\mathbf{A}}\mathcal{L}_{\mathbf{A}} + \lambda_{\mathbf{X}}\mathcal{L}_{\mathbf{X}})$$

where $\mathcal{L}_{\mathbf{C}}, \mathcal{L}_{\mathbf{A}}, \mathcal{L}_{\mathbf{X}}$ denote reconstruction losses for the curvature, adjacency, and feature matrices, while $\mathcal{L}_{\text{cls}}$ represents the cross-entropy classification loss. The anomaly score for each node is computed as a weighted sum of reconstruction errors: $\text{Score}_i = \lambda_{\mathbf{C}}|\widetilde{c}_i - c_i|_2^2 + \lambda_{\mathbf{A}}|\widetilde{a}_i - a_i|_2^2 + \lambda_{\mathbf{X}}|\widetilde{x}_i - x_i|_2^2$, where $\widetilde{c}_i, \widetilde{a}_i, \widetilde{x}_i$ are the reconstructed curvature, adjacency, and feature values, respectively. Learnable tradeoff parameters $\lambda_{\mathbf{X}}, \lambda_{\mathbf{C}}, \lambda_{\mathbf{A}}, \lambda_{\text{cls}}$ dynamically adjust loss contributions. The final anomaly detection integrates this score with classification logits, ranking anomalies based on both geometric and non-geometric deviations. An ablation study evaluating the impact of $\mathcal{L}_{\text{cls}}$ and $\mathcal{L}_{\mathbf{C}}$ is presented in Section 5.5. In the following section, we lay out the empirical results to validate the efficacy of CurvGAD.

## 5. Experimentation

### 5.1. Datasets

We evaluate CurvGAD on 10 datasets, each containing organic node-level anomalies, to assess its effectiveness across homophilic and heterophilic settings. These datasets span multiple domains, including social media, e-commerce, and financial networks. Specifically, (1) **Weibo** (Zhao et al., 2020; Liu et al., 2022), (2) **Reddit** (Kumar et al., 2019; Liu et al., 2022), (3) **Questions** (Platonov et al., 2023), and (4) **T-Social** (Tang et al., 2022) focus on identifying anomalous user behaviors on social media platforms. In the context of crowdsourcing and e-commerce, (5) **Tolokers** (Platonov et al., 2023), (6) **Amazon** (McAuley & Leskovec, 2013; Dou et al., 2020), and (7) **YelpChi** (Rayana & Akoglu, 2015; Dou et al., 2020) are used to detect fraudulent workers, reviews, and reviewers. Meanwhile, (8) **T-Finance** (Tang et al., 2022), (9) **Elliptic** (Weber et al., 2019), and (10)

| Model | Reddit | Weibo | Amazon | YelpChi | T-Finance | Elliptic | Tolokers | Questions | DGraph | T-Social | Av. Gain |
|---|---|---|---|---|---|---|---|---|---|---|---|
| GCN | $65.25_{\pm0.89}$ | $97.03_{\pm0.57}$ | $82.48_{\pm0.43}$ | $57.81_{\pm0.62}$ | $92.41_{\pm2.35}$ | $81.42_{\pm1.83}$ | $75.63_{\pm1.28}$ | $70.93_{\pm1.34}$ | $75.98_{\pm0.25}$ | $80.21_{\pm7.34}$ | 10.02 |
| GAT | $65.38_{\pm0.19}$ | $95.35_{\pm1.47}$ | $96.75_{\pm1.06}$ | $79.52_{\pm1.98}$ | $92.81_{\pm1.59}$ | $84.93_{\pm1.98}$ | $79.18_{\pm1.06}$ | $71.17_{\pm1.63}$ | $75.96_{\pm0.21}$ | $75.42_{\pm4.83}$ | 6.29 |
| GraphSAGE | $62.86_{\pm0.14}$ | $94.94_{\pm1.37}$ | $89.65_{\pm5.42}$ | $85.25_{\pm1.04}$ | $82.89_{\pm3.95}$ | $85.35_{\pm0.72}$ | $79.32_{\pm0.94}$ | $72.15_{\pm1.73}$ | $75.63_{\pm0.21}$ | $79.29_{\pm4.07}$ | 7.20 |
| BernNet | $65.82_{\pm1.75}$ | $94.96_{\pm1.56}$ | $96.21_{\pm1.43}$ | $83.07_{\pm0.61}$ | $92.81_{\pm1.78}$ | $83.18_{\pm1.49}$ | $76.95_{\pm0.65}$ | $68.82_{\pm2.98}$ | $73.25_{\pm0.17}$ | $66.85_{\pm5.84}$ | 7.74 |
| ChebyNet | $64.11_{\pm1.49}$ | $95.02_{\pm0.93}$ | $96.59_{\pm2.01}$ | $80.23_{\pm2.11}$ | $93.12_{\pm0.98}$ | $79.34_{\pm3.07}$ | $79.81_{\pm1.45}$ | $69.23_{\pm1.55}$ | $74.09_{\pm1.44}$ | $68.11_{\pm6.87}$ | 7.97 |
| HGCN | $64.13_{\pm0.98}$ | $94.23_{\pm1.86}$ | $79.94_{\pm0.93}$ | $76.46_{\pm0.62}$ | $92.32_{\pm4.18}$ | $78.66_{\pm2.22}$ | $75.78_{\pm1.01}$ | $67.16_{\pm1.74}$ | $71.27_{\pm0.25}$ | $79.43_{\pm4.22}$ | 10.00 |
| HGAT | $63.45_{\pm0.23}$ | $92.24_{\pm0.98}$ | $85.98_{\pm3.65}$ | $81.19_{\pm1.98}$ | $92.86_{\pm6.85}$ | $85.45_{\pm0.32}$ | $76.34_{\pm1.09}$ | $68.93_{\pm1.27}$ | $72.45_{\pm2.54}$ | $71.13_{\pm6.45}$ | 8.93 |
| $\kappa$GCN | $61.77_{\pm1.54}$ | $97.87_{\pm1.24}$ | $97.34_{\pm1.19}$ | $83.76_{\pm1.04}$ | $92.54_{\pm4.34}$ | $76.13_{\pm2.13}$ | $79.64_{\pm0.07}$ | $73.15_{\pm3.84}$ | $78.02_{\pm0.14}$ | $85.65_{\pm3.11}$ | 5.35 |
| $\mathcal{Q}$GCN | $66.45_{\pm1.12}$ | $97.34_{\pm0.34}$ | $96.71_{\pm5.93}$ | $83.12_{\pm1.04}$ | $92.23_{\pm5.08}$ | $85.55_{\pm2.03}$ | $81.46_{\pm0.19}$ | $71.74_{\pm2.26}$ | $78.45_{\pm1.99}$ | $73.32_{\pm4.56}$ | 5.30 |
| DCI | $66.54_{\pm3.38}$ | $94.25_{\pm1.73}$ | $94.63_{\pm0.92}$ | $77.89_{\pm7.83}$ | $86.81_{\pm4.58}$ | $82.86_{\pm1.51}$ | $75.52_{\pm0.95}$ | $69.21_{\pm1.34}$ | $74.74_{\pm0.15}$ | $80.83_{\pm6.04}$ | 7.61 |
| PCGNN | $53.25_{\pm2.19}$ | $90.27_{\pm1.51}$ | $97.31_{\pm0.82}$ | $79.76_{\pm1.57}$ | $93.34_{\pm1.02}$ | $85.89_{\pm1.85}$ | $72.83_{\pm2.07}$ | $69.91_{\pm1.43}$ | $72.08_{\pm0.35}$ | $69.27_{\pm4.49}$ | 9.54 |
| BWGNN | $65.43_{\pm4.35}$ | $97.38_{\pm0.93}$ | $97.05_{\pm0.76}$ | $84.96_{\pm0.73}$ | $94.18_{\pm0.58}$ | $85.29_{\pm1.14}$ | $80.43_{\pm0.94}$ | $71.85_{\pm2.23}$ | $76.34_{\pm0.14}$ | $82.09_{\pm5.24}$ | 4.43 |
| DOMINANT | $62.32_{\pm2.65}$ | $87.43_{\pm2.01}$ | $76.23_{\pm5.99}$ | $65.03_{\pm4.66}$ | $89.43_{\pm1.21}$ | $79.34_{\pm0.94}$ | $76.43_{\pm0.93}$ | $64.92_{\pm1.74}$ | $70.76_{\pm0.86}$ | $76.09_{\pm3.22}$ | 13.14 |
| AnomalyDAE | $65.75_{\pm1.63}$ | $93.54_{\pm1.88}$ | $91.99_{\pm0.97}$ | $81.34_{\pm0.44}$ | $91.33_{\pm0.02}$ | $83.69_{\pm0.29}$ | $77.43_{\pm1.03}$ | $62.45_{\pm0.66}$ | $78.66_{\pm1.76}$ | $82.22_{\pm0.24}$ | 7.09 |
| GADNR | $67.01_{\pm1.02}$ | $96.92_{\pm0.43}$ | $93.46_{\pm2.03}$ | $79.24_{\pm2.64}$ | $95.99_{\pm1.53}$ | $82.57_{\pm0.84}$ | $76.43_{\pm1.65}$ | $70.11_{\pm0.06}$ | $75.03_{\pm0.97}$ | $85.64_{\pm0.44}$ | 5.69 |
| ADAGAD | $66.45_{\pm0.99}$ | $98.01_{\pm0.01}$ | $96.05_{\pm1.30}$ | $85.77_{\pm1.02}$ | $92.01_{\pm2.11}$ | $86.01_{\pm0.45}$ | $80.99_{\pm0.23}$ | $71.06_{\pm1.04}$ | $75.99_{\pm2.55}$ | $84.54_{\pm2.09}$ | 4.25 |
| CurvGAD | $70.42_{\pm1.03}$ | $99.04_{\pm0.34}$ | $99.62_{\pm0.17}$ | $89.33_{\pm1.44}$ | $98.13_{\pm1.34}$ | $90.13_{\pm0.99}$ | $85.22_{\pm0.11}$ | $74.45_{\pm0.04}$ | $83.77_{\pm2.44}$ | $89.23_{\pm3.89}$ | – |
| $\Delta$ Imp. | 5.09% | 1.05% | 2.34% | 4.15% | 2.23% | 4.79% | 4.61% | 1.78% | 6.50% | 4.18% | – |

*Table 1.* CurvGAD **wins**. AUROC Score (Mean $\pm$ 95% Confidence Interval) for all baselines vs. CurvGAD. First , Second and Third best performing models for each dataset have been highlighted. **Av. Gain** shows the average absolute improvement of CurvGAD over each model across all datasets, and $\Delta$ **Imp.** indicates the percentage gain of CurvGAD over the second-best for every dataset.

**DGraph-Fin** (Huang et al., 2022) are employed to identify fraudulent users and illicit activities in financial networks. Refer to Table 6 (Appendix E.1) for dataset statistics.

## 5.2. Baselines

For a fair comparison, we evaluate CurvGAD against four types of baselines: **(a) Conventional**, including traditional models such as GCN (Kipf & Welling, 2017), GAT (Veličković et al., 2017), and SAGE (Hamilton et al., 2017). **(b) Riemannian**, comprising *(i) Constant-curvature GNNs* like HGCN (Chami et al., 2019) and HGAT (Zhang et al., 2021), and *(ii) Mixed-curvature GNNs* such as $\kappa$GCN (Bachmann et al., 2020) and $\mathcal{Q}$GCN (Xiong et al., 2022). **(c) Spectral**, including ChebyNet (Defferrard et al., 2016) and BernNet (He et al., 2021). **(d) Specialized GAD** models, including (i) *Reconstruction-based* approaches like DOMINANT (Ding et al., 2019) and AnomalyDAE (Fan et al., 2020), and (ii) *GNNs* like DCI (Wang et al., 2021), PC-GNN (Liu et al., 2021), BWGNN (Tang et al., 2022), GADNR (Roy et al., 2024), and ADAGAD (He et al., 2024).

## 5.3. Experimental Results

We conduct experiments in a transductive, supervised setting for node-level graph anomaly detection, following standard data splits where available. If splits are not provided, we adopt the strategy from (Tang et al., 2022), partitioning nodes into 40%/20%/40% for training, validation, and testing, as detailed in Table 6. To ensure robustness, we perform ten random splits per dataset and report the average performance. As per established anomaly detection benchmarks (Han et al., 2022; Liu et al., 2022), we evaluate models using

the Area Under the Receiver Operating Characteristic Curve (**AUROC**). ORC is computed with $\delta = 0.5$, distributing equal probability mass between a node and its neighbors. Given the prohibitive cost of exact ORC computation on large graphs, we employ a linear-time approximation via combinatorial bounds (Appendix B.3). The manifold *signature* is decided heuristically using the ORC distribution of the datasets (Algorithm 2, Appendix F). For optimization, we leverage the $\kappa$-stereographic product manifold implementation from Geoopt[1] and use Riemannian Adam for gradient-based learning across product manifolds. All experiments are conducted on A6000 GPUs (48GB), using a total manifold dimension of $d_{\mathcal{P}} = 48$, a learning rate of 0.01, and a filterbank comprising $\mathcal{F} = 8$ filters. Appendix E.2 enlists the hyperparameter configurations tried and we analyse the time complexity of CurvGAD in Appendix C.

## 5.4. Baseline Analysis

CurvGAD consistently outperforms all baselines across all datasets, achieving the highest AUROC scores with an improvement of upto 6.5% over the second-best model (Table 1). Significant gains are observed on T-Social (+4.18%), Elliptic (+4.79%) and Tolokers (+4.61%), all heterophilic networks, highlighting CurvGAD's ability to detect complex anomalies beyond homophily-based assumptions (addresses limitation L3). Among specialized GAD models, ADAGAD and GADNR perform well on homophilic datasets like Reddit and Weibo, ranking as second or third best in many cases. However, their performance deteriorates on heterophilic datasets, where curvature-based

[1]https://github.com/geoopt/geoopt

| AUROC | Reddit | Weibo | Amazon | YelpChi | T-Finance | Elliptic | Tolokers | Questions | DGraph | T-Social |
|---|---|---|---|---|---|---|---|---|---|---|
| CurvGAD$_{eucl}$ | 63.34±3.01 | 97.55±0.34 | 94.65±0.44 | 84.34±3.22 | 96.43±1.84 | 86.97±0.33 | 79.45±0.99 | 72.45±0.35 | 79.94±2.08 | 83.54±3.45 |
| CurvGAD$_{equi}$ | 68.75±4.03 | 98.75±0.57 | 98.34±0.67 | 85.94±1.45 | 97.87±1.23 | 87.48±0.66 | 80.54±1.01 | 71.32±0.56 | 81.37±2.56 | 83.56±2.88 |
| CurvGAD$_{invr}$ | 62.03±1.35 | 93.45±0.47 | 92.76±0.45 | 82.18±1.24 | 96.75±1.24 | 84.65±0.75 | 81.24±0.77 | 67.55±0.67 | 78.76±2.77 | 84.75±3.55 |
| CurvGAD$_{flow}$ | 66.48±3.24 | 98.77±0.84 | 97.46±0.76 | 83.26±2.34 | 96.46±1.53 | 88.46±0.34 | 74.45±0.46 | 71.74±0.35 | 80.45±3.01 | 86.34±3.69 |
| CurvGAD$_{unsp}$ | 68.34±0.99 | 98.63±0.76 | 98.01±0.55 | 87.87±2.34 | 97.22±0.86 | 88.32±0.09 | 84.01±0.58 | 73.05±0.99 | 81.05±2.01 | 88.88±1.57 |
| CurvGAD | **70.42±1.03** | **99.04±0.34** | **99.62±0.17** | **89.33±1.44** | **98.13±1.34** | **90.13±0.99** | **85.22±0.11** | **74.45±0.04** | **83.77±2.44** | **89.23±3.89** |

*Table 2.* **Ablation Study on** CurvGAD **Variants** – AUCROC Score (Mean ± 95% CI). Best performing scores per column are in **Bold**. Cell shading reflects performance degradation relative to the best variant per dataset, with a darker Gray indicating higher degradation.

| AUROC | Reddit | Weibo | Amazon | YelpChi | T-Finance | Elliptic | Tolokers | Questions | DGraph | T-Social |
|---|---|---|---|---|---|---|---|---|---|---|
| $\mathbb{H}^{24} \times \mathbb{S}^{24}$ | **70.42±1.03** | **99.04±0.24** | 96.62±0.15 | 87.56±1.66 | 96.44±1.45 | 85.45±0.95 | 83.22±0.17 | 67.56±0.45 | 75.34±2.43 | 88.45±3.05 |
| $(\mathbb{H}^{8})^2 \times (\mathbb{S}^{8})^2 \times \mathbb{E}^{16}$ | 66.43±0.98 | 97.04±0.46 | **99.62±0.19** | 85.66±1.08 | 97.56±1.34 | 88.65±0.93 | 84.56±0.18 | 71.86±0.17 | 82.46±1.87 | 84.97±3.45 |
| $\mathbb{H}^{8} \times \mathbb{S}^{8} \times \mathbb{E}^{32}$ | 68.35±1.08 | 97.01±0.12 | 95.55±0.18 | **89.33±1.44** | 96.65±1.37 | 87.36±0.45 | 82.76±0.18 | 70.56±0.07 | 80.35±2.99 | 87.56±3.35 |
| $\mathbb{H}^{16} \times (\mathbb{S}^{16})^2$ | 70.22±1.13 | 96.11±0.65 | 97.39±0.19 | 87.66±1.54 | 97.76±1.34 | 86.75±0.66 | 76.65±0.19 | 72.76±0.19 | 81.76±2.01 | **89.23±3.89** |
| $(\mathbb{H}^{16})^2 \times \mathbb{E}^{16}$ | 69.11±0.89 | 97.43±0.33 | 96.01±0.18 | 87.35±1.43 | **98.13±1.34** | 87.65±0.35 | 78.67±0.24 | 73.65±0.65 | **83.77±2.44** | 85.76±4.34 |
| $\mathbb{H}^{24} \times \mathbb{E}^{24}$ | 68.24±0.99 | 95.12±0.76 | 94.11±0.65 | 85.15±1.15 | 97.45±1.37 | 86.75±0.56 | 83.24±0.45 | **74.45±0.04** | 83.01±1.04 | 87.45±2.54 |
| $\mathbb{S}^{24} \times \mathbb{E}^{24}$ | 68.29±1.06 | 96.99±0.65 | 97.62±0.18 | 84.35±1.06 | 96.16±2.38 | 87.56±0.66 | **85.22±0.11** | 71.43±0.08 | 75.23±2.23 | 86.56±2.66 |
| $\mathbb{H}^{16} \times \mathbb{S}^{16} \times \mathbb{E}^{16}$ | 69.11±1.05 | 94.35±0.34 | 95.61±0.45 | 88.65±0.95 | 93.57±0.76 | **90.13±0.99** | 83.46±0.97 | 68.67±0.03 | 80.65±2.04 | 84.09±1.99 |

*Table 3.* **Ablation Study on Product Manifold Signatures** – AUCROC Score (Mean ± 95% CI). Best performing *signatures* are in **Bold**. Cell shading reflects performance degradation relative to the best signature per dataset, with darker Gray indicating higher degradation.

models like $\kappa$GCN and $\mathcal{Q}$GCN perform better, suggesting that explicitly modeling curvature aids anomaly detection in such settings. Reconstruction-based approaches like DOMINANT and AnomalyDAE perform worse than dedicated GAD approaches, indicating that pure structure and attribute reconstruction alone may be insufficient for anomaly detection. These models fail to capture deeper geometric irregularities, reinforcing the necessity of integrating curvature information. HGCN and HGAT, despite leveraging hyperbolic geometry, underperform compared to CurvGAD, as they operate on a constant curvature space, limiting their adaptability to datasets with mixed geometric properties.

## 5.5. Ablation Study

To assess the contribution of individual components in CurvGAD, we evaluate multiple ablations (Tables 2, 3).

**(a) Euclidean-only variant** (CurvGAD$_{eucl}$). This model learns purely in Euclidean space, discarding mixed-curvature embeddings. The performance drop ($\sim 5\%$) highlights the necessity of mixed-curvature modeling (addresses L1). Notably, degradation is more pronounced on curvature-sensitive datasets like Reddit ($-7.08\%$) and T-Social ($-5.69\%$) but is minimal on T-Finance ($-1.7\%$), where Ollivier-Ricci curvatures are predominantly zero.

**(b) Without Ricci Flow** (CurvGAD$_{flow}$). Removing Ricci flow prevents curvature-invariant reconstruction, degrading performance on curvature-sensitive graphs ($-10.77\%$ on Tolokers, $-3.94\%$ on Reddit). This confirms Ricci flow's role in stabilizing curvature variations and enhancing structural and attribute reconstruction. Observe that in Weibo, the

drop is minimal ($-0.27\%$), which can be attributed to the low curvature variance in the ORC distribution of Weibo.

**(c) Manifold signature analysis**. The optimal curvature composition varies across datasets (Table 3). Reddit and Questions favor hyperbolic embeddings, while Amazon and YelpChi perform better with increased Euclidean capacity. This validates the necessity of mixed-curvature manifolds for adapting to diverse graph structures. To quantify the contribution of individual geometry types, we further conduct ablations by removing each component in turn, e.g. $\mathbb{H}^{24} \times \mathbb{S}^{24}$ ($\mathbb{E}$ removed) and $\mathbb{S}^{24} \times \mathbb{E}^{24}$ ($\mathbb{H}$ removed).

**(d) Curvature-invariant pipeline only** (CurvGAD$_{invr}$). This variant omits curvature reconstruction, relying solely on structural and attribute reconstruction (removes $\mathcal{L}_\mathbf{C}$). While it still surpasses most baselines, performance declines significantly ($-8.39\%$ on Reddit, $-6.86\%$ on Amazon), confirming that structure and attributes alone are insufficient for robust anomaly detection (addresses L2).

**(e) Curvature-equivariant pipeline only** (CurvGAD$_{equi}$). This model detects only geometric anomalies, omitting curvature-invariant reconstruction (removes $\mathcal{L}_\mathbf{A}, \mathcal{L}_\mathbf{X}$). It outperforms CurvGAD$_{invr}$ ($+6.72\%$ on Reddit, $+4.89\%$ on Amazon), highlighting how effective the curvature-equivariant pipeline is in itself. Both pipelines compliment each other, and achieve the best results together.

**(f) Unsupervised variant** (CurvGAD$_{unsp}$). Removing the supervised classification loss $\mathcal{L}_{cls}$ results in a minor performance drop ( 1.5% avg.), demonstrating that CurvGAD remains effective even in an unsupervised setting, making it adaptable to real-world scenarios with limited labels.

These results validate the independent contributions of each component, showing that curvature-awareness, curvature-invariance, and Ricci flow collectively enhance anomaly detection in both homophilic and heterophilic graphs.

## 6. Conclusion

In this paper, we propose CurvGAD, the **first** approach to introduce the notion of curvature-based *geometric* anomalies and to model them in graph anomaly detection through a mixed-curvature perspective. Our dual-pipeline architecture integrates curvature-equivariant and curvature-invariant reconstruction, capturing diverse anomalies across homophilic and heterophilic networks. CurvGAD disentangles geometric, structural, and attribute-based anomalies, making it interpretable. Extensive experiments across 10 benchmarking datasets demonstrate the superiority of CurvGAD over existing baselines. These results establish curvature as a fundamental tool for GAD and pave the way for future advancements in geometry-aware graph learning while offering new avenues for exploring *geometric* anomalies in topologically complex networks.

## Impact Statement

The goal of this work is to offer a general-purpose anomaly detection framework that has broad applicability and theoretical foundations. Although we do not anticipate any immediate ethical issues specific to our study, careful implementation in practical contexts is required for security and fairness in automated anomaly detection systems.

The potential uses of CurvGAD include fraud detection, cybersecurity, financial risk analysis, and misinformation detection, even though it was created as a foundational research contribution to machine learning and graph representation learning. Due to the high stakes involved in decision-making, these fields raise ethical questions about prejudice, justice, and openness. Biases in input data may be exacerbated in anomaly identification results because our approach depends on graph topology and node-level features. This is especially true in financial or social network investigations where marginalized groups may be disproportionately identified as outliers. Therefore, it is essential to make sure that datasets are selected using preprocessing that considers fairness and that model outputs are carefully assessed.

Additionally, CurvGAD makes GAD techniques more scalable, which could make them an effective tool for extensive monitoring systems. Although advantageous for anti-fraud and cybersecurity efforts, malicious actors might try to alter graph architecture in order to avoid detection. Robustness against adversarial perturbations in curvature-aware models should be investigated in future studies. Our method improves interpretability by separating geometric and non-geometric anomalies, which is important in high-stakes applications where decision-making requires model explanations. In scientific domains including network science, biology, neurology, and transportation networks, where curvature-driven representations may aid in identifying anomalous patterns in actual systems, the knowledge gained via CurvGAD might be advantageous.

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

# A. Appendix

## A.1. Notation Table

| Notation | Reference |
|---|---|
| $\mathcal{G} = (\mathcal{V}, \mathcal{E}, \mathbf{A})$ | A graph $\mathcal{G}$ with set of edges $\mathcal{E}$ and vertices (nodes) $\mathcal{V}$ |
| $\mathcal{M}; \mathcal{M}_p$ | A smooth manifold; $p^{th}$ component manifold in the product manifold ($\mathcal{M}_p \in \{\mathbb{E}, \mathbb{H}, \mathbb{S}\}$) |
| $\kappa_p, d_p$ | Curvature and dimension of $p^{th}$ manifold component |
| $\exp_0^\kappa : \mathbb{R}^{d_\mathcal{P}} \to \mathcal{M}$ | Exponential map, to map from tangent plane (Euclidean) to the product manifold |
| $\log_0^\kappa : \mathcal{M} \to \mathbb{R}^{d_\mathcal{P}}$ | Logarithmic map, to map from the manifold to the tangent plane |
| $\mathbb{H}; \mathbb{S}; \mathbb{E}$ | Hyperbolic manifold; Spherical manifold; Euclidean manifold |
| $\mathbb{P}^{d_\mathcal{P}}$ | Product manifold of dimension $d_\mathcal{P}$ |
| $\kappa; \widetilde{\kappa}$ | Continous manifold curvature; Discrete Ollivier-Ricci curvature (ORC) |
| $\widetilde{\kappa}_{xy}; \widetilde{\kappa}_{xy}^{(t)}$ | ORC of edge $\{x, y\}$, ORC of edge $\{x, y\}$ at $t^{th}$ step of Ricci flow algorithm |
| $m_x^\delta$ | Probability mass assigned to node $x$ for ORC computation |
| $\delta$ | Probability of the mass retained v/s distributed among neighbour nodes for ORC Computation |
| $\mathcal{N}(x)$ | Neighbourhood set of node $x$ |
| $x \sim y$ | This implies that $x$ and $y$ are adjacent nodes |
| $\mathbf{W}_1(.)$ | Wasserstein-1 (Earth mover's) distance |
| $d_\mathcal{G}(x, y)$ | Shortest path (graph distance) between nodes $x$ and $y$ on graph $\mathcal{G}$ |
| $\mathcal{M}_p^{\kappa_p, d_p}$ | Constant-curvature manifold with dimension $d_p$ and curvature $\kappa_p$. $\mathcal{M}_p \in \{\mathbb{H}, \mathbb{S}, \mathbb{E}\}$ |
| $\mathbf{L}_\mathbb{P}$ | Discrete Laplace-Beltrami operator |
| $\mathbf{U}_\mathbb{P} = \left[\{\mathbf{u}_l\}_{l=0}^{n-1}\right]$ | Set of orthonormal eigenvectors of $\mathbf{L}_\mathbb{P}$ |
| $\left[\{\lambda_l\}_{l=0}^{n-1}\right] \in \mathbb{R}^n$ | Ordered real nonnegative eigenvalues associated with $\mathbf{U}_\mathbb{P}$ |
| $\epsilon$ | Ollivier-Ricci flow step size |
| $\mathbf{w}_{xy}^{(t)}$ | Weight of the edge $xy$ at the $t^{th}$ iteration in Ricci flow algorithm. |
| $d_\mathcal{X}$ | Input graph feature dimension |
| $d_\mathcal{P}$ | Total dimension of the product manifold |
| $\mathbf{X} \in \mathbb{R}^{n \times d_\mathcal{X}}$ | Input feature matrix |
| $\oplus_\kappa$ | *Mobius* addition |
| $\otimes_\kappa$ | $\kappa$-right-matrix-multiplication |
| $\boxtimes_\kappa$ | $\kappa$-left-matrix-multiplication |
| $\mathbf{n}_x$ | Final (latent) node representation for node $x$ after encoder |
| $\mathbf{A}; \widetilde{\mathbf{A}}$ | Initial graph adjacency matrix, Reconstructed adjacency matrix |
| $\mathbf{X}; \widetilde{\mathbf{X}}$ | Initial graph feature matrix, Reconstructed feature matrix |
| $\mathbf{C}; \widetilde{\mathbf{C}}$ | Initial edge curvature matrix, Reconstructed curvature matrix |
| $\mathcal{L}_\mathbf{C}$ | Curvature decoder reconstruction loss |
| $\mathcal{L}_\mathbf{A}$ | Structure decoder reconstruction loss |
| $\mathcal{L}_\mathbf{X}$ | Feature decoder reconstruction loss |
| $\mathcal{L}_\text{cls}$ | Binary crossentropy supervised classification loss |
| $\lambda_\text{cls}$ | Weight of classification loss in total loss |
| $\mathcal{D}_{\mathcal{M}_p}$ | Geodesic distance over $p^{th}$ manifold component |
| $\mathcal{D}_\mathcal{M}(\mathbf{n}_x, \mathbf{n}_y)$ | Aggregated distance on the product manifold (i.e. sum over all component manifolds) |
| $\mathcal{K}(\mathbf{n}_x, \mathbf{n}_y)$ | Gaussian kernel used in curvature decoder over node embeddings $\mathbf{n}_x, \mathbf{n}_y$ |
| $\beta_p$ | Learnable attention weight for $p^{th}$ component manifold |
| $\alpha_f$ | Learnable attention weight for $f^{th}$ filter in the filterbank |
| $\mathcal{F}; f$ | Total number of filters in the filterbank; Used to denote the $f^{th}$ filter |
| $\mathcal{P}; p$ | Total number of components in the product manifold; Used to denote the $p^{th}$ component |
| $\lambda_\mathbf{X}, \lambda_\mathbf{C}, \lambda_\mathbf{A}, \lambda_\text{cls}$ | Learnable trade-off parameters for loss components |
| $\sigma(.)$ | Sigmoid activation function |
| $\Delta$ | Stopping criterion threshold for Ollivier-Ricci flow |
| $\psi$ | Graph filter operator |
| $f_\theta(.)$ | Neural network (MLP) that generates hidden state Euclidean features of dimension $d_\mathcal{P}$ |
| $\mathbf{x}$ | A graph signal $\mathbf{x} : \mathcal{V} \to \mathbb{R}$ |
| $\gamma$ | Kernel width hyperparameter for gaussian kernel (curvature decoder) |
| $\tau$ | Sensitivity parameter (learnable) for gaussian kernel (curvature decoder) |
| $\mathbf{Z}_{\mathcal{M}_p^{\kappa_p, d_p}}^{(f)}$ | Intermediate node representation from $f^{th}$ filter |
| $T_l(x)$ | Chebyshev polynomial of order $l$ |
| $\Lambda_\mathbb{P}$ | Diagonal matrix of eigenvalues |
| $\phi_f$ | Learnable weights for filters, for the $f^{th}$ recursive level in Chebyshev encoder. |

# B. More on Preliminaries

### B.1. Product Manifolds

Consider a set of $\mathcal{P}$ smooth manifolds $\mathcal{M}_1, \mathcal{M}_2, \ldots, \mathcal{M}_\mathcal{P}$, each equipped with its own Riemannian structure. The **product manifold** $\mathbb{P}$ is formed as their Cartesian product:

$$\mathbb{P} = \mathcal{M}_1 \times \mathcal{M}_2 \times \cdots \times \mathcal{M}_\mathcal{P}. \tag{7}$$

This construction enables the representation of a point $\mathbf{x} \in \mathbb{P}$ as a tuple:

$$\mathbf{x} = (x_1, x_2, \ldots, x_\mathcal{P}), \tag{8}$$

where each $x_p$ corresponds to a coordinate in the individual component manifold $\mathcal{M}_p$.

■ **Tangent Space and Geodesic Movement**. For any given $\mathbf{x} \in \mathbb{P}$, the associated tangent space is given by:

$$\mathbf{T_x}\mathbb{P} = \mathbf{T}_{x_1}\mathcal{M}_1 \times \mathbf{T}_{x_2}\mathcal{M}_2 \times \cdots \times \mathbf{T}_{x_\mathcal{P}}\mathcal{M}_\mathcal{P}. \tag{9}$$

A tangent vector $\mathbf{v} \in \mathbf{T_x}\mathbb{P}$ is similarly represented as:

$$\mathbf{v} = (v_1, v_2, \ldots, v_\mathcal{P}), \tag{10}$$

where each $v_p \in \mathbf{T}_{x_p}\mathcal{M}_p$ represents the directional component along the corresponding manifold $\mathcal{M}_p$.

Since optimization over Riemannian manifolds requires movement along the manifold surface rather than within the tangent space, we employ the **exponential map**, which maps a tangent vector $\mathbf{v}$ at $\mathbf{x}$ back onto the manifold:

$$\mathbf{exp_x} : \mathbf{T_x}\mathbb{P} \to \mathbb{P}. \tag{11}$$

For a product manifold, this mapping decomposes across individual components, leading to:

$$\mathbf{exp_x}(\mathbf{v}) = (\mathbf{exp}_{x_1}(v_1), \mathbf{exp}_{x_2}(v_2), \ldots, \mathbf{exp}_{x_\mathcal{P}}(v_\mathcal{P})). \tag{12}$$

This property allows optimization steps to be performed independently along each manifold component, significantly simplifying Riemannian optimization in mixed-curvature spaces. We use the **Riemannian Adam** optimizer from Geoopt (https://github.com/geoopt/geoopt) Library for all our experimentation.

■ **Curvature and Manifold Selection**. The choice of component manifolds in $\mathbb{P}$ significantly influences model performance. Hyperbolic spaces ($\mathbb{H}$) are well-suited for hierarchical structures, spherical spaces ($\mathbb{S}$) capture cyclical relations, while Euclidean spaces ($\mathbb{E}$) are ideal for line graphs (and provide a standard representation). The flexibility of mixed-curvature embeddings allows for adaptive modeling across datasets with diverse underlying geometries.

*Remark* B.1. CurvGAD heuristically determines the optimal mixed-curvature manifold configuration (Algorithm 2, Appendix F) for a given dataset, ensuring that embeddings conform to the dataset's intrinsic geometric properties.

*Remark* B.2. The curvatures of the manifold components are dynamically **learnt** during the training process.

By leveraging a product of Riemannian manifolds, our approach effectively models complex graphs where curvature varies locally, leading to superior anomaly detection across diverse graph topologies.

### B.2. $\kappa-$Stereographic Model

To effectively model mixed-curvature embeddings, we adopt the $\kappa$-stereographic model (Bachmann et al., 2020) instead of adopting separate formulations for hyperbolic and spherical spaces. This unified framework provides a consistent mathematical formulation across varying curvature regimes, enabling smooth interpolation between hyperbolic ($\kappa < 0$), spherical ($\kappa > 0$), and Euclidean ($\kappa = 0$) geometries. Efficient algebraic operations are essential for optimizing over non-Euclidean spaces, and the $\kappa$-stereographic model offers closed-form expressions for key Riemannian operations.

■ **Manifold Definition**. The $\kappa$-stereographic manifold of curvature $\kappa$ and dimension $d$ is defined as:

$$\mathcal{M}^{\kappa,d} = \left\{ \mathbf{z} \in \mathbb{R}^d \mid -\kappa\|\mathbf{z}\|_2^2 < 1 \right\}, \tag{13}$$

| Operation | Formalism in $\mathbb{E}^d$ | Unified formalism in $\kappa$-stereographic model ($\mathbb{H}^d$/$\mathbb{S}^d$) |
|---|---|---|
| Distance Metric | $d_{\mathcal{M}}^\kappa(\mathbf{x}, \mathbf{y}) = \|\mathbf{x} - \mathbf{y}\|_2$ | $d_{\mathcal{M}}^\kappa(\mathbf{x}, \mathbf{y}) = \frac{2}{\sqrt{|\kappa|}} \tan_\kappa^{-1}\left(\sqrt{|\kappa|}\, \|-\mathbf{x} \oplus_\kappa \mathbf{y}\|_2\right)$ |
| Exp. Map | $\exp_\mathbf{x}^\kappa(\mathbf{v}) = \mathbf{x} + \mathbf{v}$ | $\exp_\mathbf{x}^\kappa(\mathbf{v}) = \mathbf{x} \oplus_\kappa \left(\tan_\kappa\left(\sqrt{|\kappa|}\frac{\lambda_\mathbf{x}^\kappa\|\mathbf{v}\|_2}{2}\right)\frac{\mathbf{v}}{\sqrt{|\kappa|}\|\mathbf{v}\|_2}\right)$ |
| Log. Map | $\log_\mathbf{x}^\kappa(\mathbf{y}) = \mathbf{x} - \mathbf{y}$ | $\log_\mathbf{x}^\kappa(\mathbf{y}) = \frac{2}{\sqrt{|\kappa|}\lambda_\mathbf{x}^\kappa} \tan_\kappa^{-1}\left(\sqrt{|\kappa|}\, \|-\mathbf{x} \oplus_\kappa \mathbf{y}\|_2\right)\frac{-\mathbf{x}\oplus_\kappa\mathbf{y}}{\|-\mathbf{x}\oplus_\kappa\mathbf{y}\|_2}$ |
| Addition | $\mathbf{x} \oplus_\kappa \mathbf{y} = \mathbf{x} + \mathbf{y}$ | $\mathbf{x} \oplus_\kappa \mathbf{y} = \frac{\left(1+2\kappa\mathbf{x}^T\mathbf{y}+K\|\mathbf{y}\|^2\right)\mathbf{x}+\left(1-\kappa\|\mathbf{x}\|^2\right)\mathbf{y}}{1+2\kappa\mathbf{x}^T\mathbf{y}+\kappa^2\|\mathbf{x}\|^2\|\mathbf{v}\|^2}$ |

*Table 4.* Operations in Hyperbolic $\mathbb{H}^d$, Spherical $\mathbb{S}^d$ and Euclidean space $\mathbb{E}^d$.

where $\mathbf{z} \in \mathbb{R}^d$ represents a point in the stereographic space, and the curvature parameter $\kappa$ governs the underlying geometry: $\kappa < 0$ corresponds to hyperbolic manifold ($\mathbb{H}^d$), $\kappa > 0$ corresponds to a spherical manifold ($\mathbb{S}^d$) and $\kappa = 0$ reduces to Euclidean space ($\mathbb{E}^d$). The Riemannian metric tensor associated with $\mathcal{M}^{\kappa,d}$ at a point $\mathbf{z}$ is:

$$g_\mathbf{z}^\kappa = (\lambda_\mathbf{z}^\kappa)^2 \mathbf{I}, \tag{14}$$

where the conformal factor $\lambda_\mathbf{z}^\kappa$ is defined as:

$$\lambda_\mathbf{z}^\kappa = 2\left(1 + \kappa\|\mathbf{z}\|_2^2\right)^{-1}. \tag{15}$$

This ensures a smooth transition between different curvature settings.

■ **Algebraic Operations**. To perform optimization and inference on the manifold, we require efficient algebraic formulations for fundamental operations such as vector addition, scalar multiplication, and distance computation. Table 4 summarizes the key operations under the $\kappa$-stereographic model.

### B.2.1. $\kappa$-RIGHT-MATRIX-MULTIPLICATION (BACHMANN ET AL., 2020)

For a matrix $\mathbf{X} \in \mathbb{R}^{n \times d}$ carrying $\kappa$-stereographic embeddings across its rows, and weights represented by $\mathbf{W} \in \mathbb{R}^{d \times e}$, the operation of Euclidean right multiplication can be decomposed into individual rows as $(\mathbf{XW})_{i\bullet} = \mathbf{X}_{i\bullet}\mathbf{W}$. Consequently, the $\kappa$-right-matrix-multiplication is defined in the same row-wise fashion as

$$(\mathbf{X} \otimes_\kappa \mathbf{W})_{i\bullet} = \exp_0^\kappa\left((\log_0^\kappa(\mathbf{X})\mathbf{W})_{i\bullet}\right) \tag{16}$$

$$= \tan_\kappa\left(\alpha_i \tan_\kappa^{-1}(\|\mathbf{X}_{\bullet i}\|)\right)\frac{(\mathbf{XW})_{i\bullet}}{\|(\mathbf{XW})_{i\bullet}\|} \tag{17}$$

where $\alpha_i = \frac{\|(\mathbf{XW})_{i\bullet}\|}{\|\mathbf{X}_{i\bullet}\|}$ and $\exp_0^\kappa$ and $\log_0^\kappa$ denote the exponential and logarithmic map in the $\kappa$-stereographic model.

### B.3. Ollivier-Ricci Curvature (ORC)

In an unweighted graph, the set of nodes adjacent to a given node $x$, denoted by $\mathcal{N}(x)$, is represented through a probability distribution following a lazy random walk model (Lin et al., 2011). This distribution is defined by the equation:

$$m_z^\delta(x) = \begin{cases} \delta, & \text{if } z = x, \\ \frac{1-\delta}{|\mathcal{N}(x)|}, & \text{if } z \in \mathcal{N}(x), \\ 0, & \text{otherwise.} \end{cases} \tag{18}$$

The parameter $\delta$ determines the probability of staying at the current node, with the rest of the probability $(1 - \delta)$ being evenly allocated among its neighbors. This approach links ORC with lazy random walks, affecting the interaction between local exploration and node revisitation. In our study, we set $\delta = 0.5$, resulting in an equal division of probability between the node and its neighbors, creating a balance between *breadth-first* and *depth-first* search methodologies. The value of $\delta$ is significant and varies based on the graph's structure. A lower $\delta$ value promotes more exploration within the local neighborhood, whereas a higher $\delta$ fosters node revisits, supporting the "lazy" aspect of the walk. For the experiments, $\delta = 0.5$ was chosen to ensure an equal division of probability between the node and its adjacent nodes.

# C. Time Complexity Analysis

## C.1. Preprocessing Complexity

### C.1.1. COMPUTATIONAL CONSIDERATIONS FOR ORC

A crucial preprocessing step in CurvGAD is the computation of the **Ollivier-Ricci Curvature (ORC)** and subsequent **Ricci Flow** regularization. These computations serve as the foundation for both the curvature-invariant and curvature-equivariant pipelines, making their efficiency paramount to overall scalability. We precompute both these to save computation time as they are used in both the CurvGAD pipelines.

Computing ORC involves determining (in this case, approximating) the *Wasserstein-1 distance* ($\mathbf{W}_1$) between the probability distributions of neighboring nodes. In discrete settings, this requires solving an optimal transport problem. The standard approach relies on the *Hungarian algorithm* (Kuhn, 1955), which operates in $\mathcal{O}(d^3)$ time per node, making it computationally prohibitive for large graphs.

To mitigate this cost, the *Sinkhorn algorithm* (Sinkhorn & Knopp, 1967) is often used as a relaxation, reducing complexity to $\mathcal{O}(d^2)$ per node. However, this still remains impractical for million-scale graphs. **In this work, we adopt a linear-time approximation for ORC**, allowing CurvGAD to scale efficiently while retaining interpretability.

### C.1.2. APPROXIMATING ORC IN LINEAR TIME

Instead of directly solving the optimal transport problem, we employ a *combinatorial approximation* for ORC, inspired by Tian et al. (2023). This approach estimates Wasserstein distances using local structural properties, significantly reducing computational overhead. The approximation is derived from the analytical curvature bounds proposed by Jost & Liu (2014), which leverage local connectivity features such as *node degrees* and *triangle counts*. Let $\#(x, y)$ represent the number of triangles involving the edge $(x, y)$, and define $a \wedge b = \min(a, b)$, $a \vee b = \max(a, b)$, where $d_x$ denotes the degree of node $x$. The Ollivier-Ricci curvature for edge $(x, y)$ is then bounded as follows:

**Theorem C.1** (Jost & Liu (2014))**.** *For an unweighted graph, the Ollivier-Ricci curvature of edge $e = (x, y)$ satisfies:*

1. *Lower bound:*

$$
\widetilde{\kappa}(x, y) \geq -\left(1 - \frac{1}{d_x} - \frac{1}{d_y} - \frac{\#(x, y)}{d_x \wedge d_y}\right)_+ \tag{19}
$$
$$
-\left(1 - \frac{1}{d_x} - \frac{1}{d_y} - \frac{\#(x, y)}{d_x \vee d_y}\right)_+ + \frac{\#(x, y)}{d_x \vee d_y}.
$$

2. *Upper bound:*

$$
\widetilde{\kappa}(x, y) \leq \frac{\#(x, y)}{d_x \vee d_y}. \tag{20}
$$

The curvature of an edge can then be efficiently approximated as:

$$
\widehat{\kappa}(x, y) := \frac{1}{2}\left(\kappa^{\text{upper}}(x, y) + \kappa^{\text{lower}}(x, y)\right). \tag{21}
$$

■ **Computational Complexity.** This approximation scheme runs in **linear time** $\mathcal{O}(|\mathcal{E}|)$, where $|\mathcal{E}|$ is the number of edges. Since each edge's curvature is estimated using only local information (node degrees and triangles), the process is inherently **parallelizable** across edges, making it well-suited for large-scale graphs.

### C.1.3. RICCI FLOW COMPLEXITY

Ricci Flow is an iterative process where edge weights are adjusted according to the ORC values, smoothing the curvature distribution over time. Each iteration requires one full computation of ORC, making the total complexity: $\mathcal{O}(I|\mathcal{E}|)$, where $I$ is the number of iterations required for convergence. Empirically, across the datasets used in this study, Ricci Flow converges in 12–13 iterations on average. Since $I$ is much smaller than the total number of edges ($I \ll |\mathcal{E}|$), the process remains linear in complexity:, i.e. $\mathcal{O}(I|\mathcal{E}|) \approx \mathcal{O}(|\mathcal{E}|)$ as $I$ is fixed across datasets.

### C.1.4. SUMMARY OF PREPROCESSING COMPLEXITY

- Ollivier-Ricci Curvature (ORC): $\mathcal{O}(|\mathcal{E}|)$ (linear time)

- Ricci Flow Regularization: $\mathcal{O}(|\mathcal{E}|)$ (empirically bounded by a constant number of iterations)

- Total Preprocessing Complexity: $\mathcal{O}(|\mathcal{E}|)$ (**parallelizable** and scalable)

## C.2. Runtime Comparison

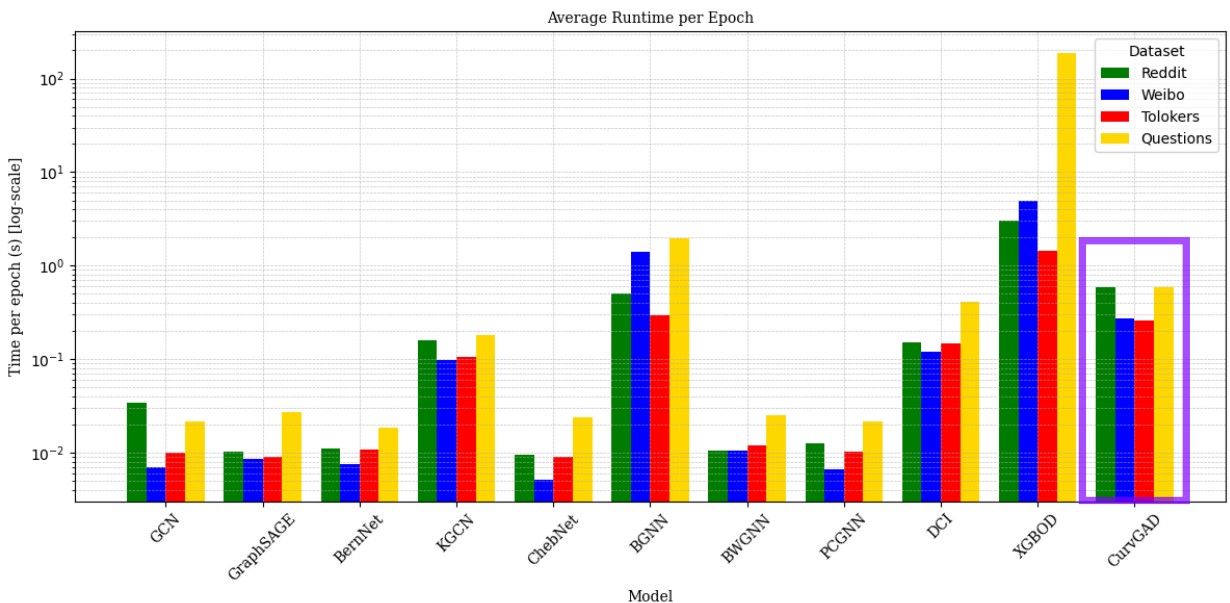

*Figure 4.* **Runtime comparision of** CurvGAD. Average runtime per epoch (training + inference) in *seconds* (log scale) on Reddit (Liu et al., 2022), Weibo (Zhao et al., 2020), Tolokers (Platonov et al., 2023), and Questions (Platonov et al., 2023) datasets, for select baselines.

| Dataset | GCN | GraphSAGE | GAT | BernNet | KGCN | ChebNet | BGNN | BWGNN | PCGNN | DCI | GHRN | XGBOD | CurvGAD |
|---|---|---|---|---|---|---|---|---|---|---|---|---|---|
| **Reddit** | 34.37 | 10.19 | 17.55 | 11.27 | 158.45 | 9.52 | 507.41 | 10.52 | 12.50 | 150.61 | 11.70 | 3050.66 | 585.22 |
| **Weibo** | 6.95 | 8.58 | 10.63 | 7.65 | 99.34 | 5.13 | 1415.59 | 10.63 | 6.74 | 121.77 | 11.06 | 4889.16 | 276.70 |
| **Tolokers** | 10.12 | 9.02 | 12.17 | 10.80 | 105.23 | 9.16 | 295.89 | 11.94 | 10.37 | 149.58 | 14.08 | 1432.76 | 260.30 |
| **Questions** | 21.56 | 27.12 | 22.85 | 18.53 | 183.53 | 23.77 | 1972.41 | 25.41 | 21.44 | 408.37 | 26.55 | 188192.5 | 589.41 |

*Table 5.* **Runtime Comparison (milliseconds)** of CurvGAD and baseline models across multiple datasets. This table shows the exact numbers used in Figure 4. The results are shown for CurvGAD model with $\mathcal{F} = 8$ filters and product manifold $(\mathbb{H}^8)^2 \times \mathbb{S}^{16} \times \mathbb{E}^{16}$.

### C.2.1. RUNTIME ANALYSIS

The runtime comparison in Table C.2 provides key insights into the computational efficiency of CurvGAD relative to baseline methods. Our results report the execution time in milliseconds for CurvGAD configured with $\mathcal{F} = 8$ spectral filters and a mixed-curvature product manifold $(\mathbb{H}^8)^2 \times \mathbb{S}^{16} \times \mathbb{E}^{16}$. Despite leveraging multiple curvature spaces and graph filters for representation learning, CurvGAD demonstrates reasonable computational efficiency. For reference, when analysing Figure 4, GCN and BerNet can be thought of as using just one graph filter and KGCN can be considered as one manifold for learning representations (when compared to CurvGAD with 8 filters and 4 manifolds).

- **Scalability with mixed-curvature manifolds.** Unlike KGCN, which operates on a single curvature space, CurvGAD effectively integrates four manifold components. Nevertheless, its runtime remains within an acceptable range, significantly outperforming KGCN across all datasets. For instance, on Reddit, KGCN requires 158.45ms, while CurvGAD completes execution in 585.22ms, despite utilizing a far more expressive model.

- **Efficiency compared to BGNN and XGBOD.** CurvGAD remains substantially faster than models like BGNN and XGBOD. On the Weibo dataset, BGNN exhibits an extreme runtime of 1415.59ms, whereas CurvGAD operates at only 276.70ms. Similarly, on Tolokers, BGNN requires 295.89ms, compared to 260.30ms for CurvGAD, making it significantly more efficient despite handling curvature-sensitive graph learning.

- **Reasonable overhead given model complexity.** The additional computational cost incurred by CurvGAD is expected due to its curvature-based learning framework, which integrates Ricci flow, manifold embeddings, and spectral filtering. However, the runtime remains competitive, ensuring that the benefits of richer geometric representations do not come at an impractical computational expense.

Overall, CurvGAD balances computational efficiency and model expressiveness, outperforming single-curvature models while maintaining a significantly lower runtime than complex baselines like BGNN and XGBOD. Its efficiency relative to KGCN and its comparable performance to DCI suggest that the inclusion of curvature-based representations does not drastically hinder scalability, making it a viable choice for large-scale graph anomaly detection tasks.

## D. Arcitectural Details

### D.1. Definition 4.1: Intuition and Derivation

The curvature decoder in Section 4.1.2 reconstructs the Ollivier-Ricci curvature (ORC) $\mathbf{C}_{xy}$ between nodes $x$ and $y$ using their latent embeddings $\mathbf{n}_x, \mathbf{n}_y \in \mathbb{P}^{d_{\mathcal{M}}}$. The key idea is to define a *Gaussian kernel* on the geodesic distance within the product manifold to capture the intrinsic curvature information. In this section, we derive the decoder formulation and provide mathematical intuition.

**Motivation: Distance-based curvature estimation**. Curvature measures the deviation of a space from being flat. The Ollivier-Ricci curvature is based on the Wasserstein distance between local probability distributions around connected nodes $x$ and $y$. Intuitively, negatively curved spaces (e.g., hyperbolic manifolds) exhibit greater spread between transported probability masses, whereas positively curved spaces (e.g., spherical manifolds) result in concentrated transport.

Given that ORC is inherently tied to how distances between points behave in different curvature regimes, it is natural to model curvature reconstruction as a function of the geodesic distance $\mathcal{D}_{\mathcal{M}}(\mathbf{n}_x, \mathbf{n}_y)$. A widely adopted approach in Riemannian learning is to employ a Gaussian kernel, which serves as a smooth and flexible mechanism to relate distances to similarity metrics.

**Geodesic Distance on the Product Manifold**. For a product manifold $\mathbb{P}$ composed of $\mathcal{M}$ individual manifold components, the squared geodesic distance between node embeddings $\mathbf{n}_x$ and $\mathbf{n}_y$ is computed as:

$$\mathcal{D}_{\mathcal{M}}(\mathbf{n}_x, \mathbf{n}_y)^2 = \sum_{p=1}^{\mathcal{M}} \mathcal{D}_{\mathcal{M}_p}(n_x, n_y)^2. \tag{22}$$

This formulation ensures that distances are computed per manifold component and then aggregated across all curvature spaces.

**Gaussian Kernel Formulation**. To reconstruct the curvature, we define a Gaussian kernel based on the computed geodesic distance:

$$\mathcal{K}(\mathbf{n}_x, \mathbf{n}_y) = \exp\left(-\gamma \frac{\mathcal{D}_{\mathcal{M}}(\mathbf{n}_x, \mathbf{n}_y)^2}{\tau^2}\right), \tag{23}$$

where $\gamma$ controls the width of the kernel and $\tau$ is a trainable sensitivity parameter that adapts to the curvature distribution.

*Intuition.* The kernel output is close to 1 for small distances (nodes close in latent space) and approaches 0 for large distances (nodes far apart). This behavior aligns with the principle that lower curvature corresponds to more spread-out distances in latent space.

**Sigmoid transformation for curvature prediction**. The predicted curvature $\widetilde{\mathbf{C}}_{xy}$ is derived from the kernel output via a sigmoid transformation:

$$\widetilde{\mathbf{C}}_{xy} = 2 \cdot \texttt{sigmoid}\left(1 - \mathcal{K}(\mathbf{n}_x, \mathbf{n}_y)\right) - 1. \tag{24}$$

*Justification:* The sigmoid function normalizes the kernel output to the $[0, 1]$ range, and the affine transformation maps the values into the $[-1, 1]$ range, consistent with the properties of Ollivier-Ricci curvature.

**Optimization via reconstruction loss**. To ensure accurate curvature reconstruction, we optimize the decoder by minimizing the Frobenius norm loss:

$$\mathcal{L}_{\mathbf{C}} = \left\| \widetilde{\mathbf{C}} - \mathbf{C} \right\|_F^2, \tag{25}$$

where $\widetilde{\mathbf{C}}$ denotes the predicted curvature matrix and $\mathbf{C}$ is the true ORC matrix.

**Summary:** The decoder exploits the structure of the product manifold by computing geodesic distances, mapping them through a Gaussian kernel, and applying a transformation to ensure proper curvature scaling. This approach provides an effective means of capturing geometric distortions while maintaining computational efficiency.

# E. Experimental Details

### E.1. Dataset Statistics

| | #Nodes | #Edges | #Feat. | Anomaly | Train | Relationship |
|---|---|---|---|---|---|---|
| Reddit (Kumar et al., 2019; Liu et al., 2022) | 10,984 | 168,016 | 64 | 3.3% | 40% | Under Same Post |
| Weibo (Zhao et al., 2020; Liu et al., 2022) | 8,405 | 407,963 | 400 | 10.3% | 40% | Under Same Hashtag |
| Amazon (McAuley & Leskovec, 2013; Dou et al., 2020) | 11,944 | 4,398,392 | 25 | 9.5% | 70% | Review Correlation |
| YelpChi (Rayana & Akoglu, 2015; Dou et al., 2020) | 45,954 | 3,846,979 | 32 | 14.5% | 70% | Reviewer Interaction |
| Tolokers (Platonov et al., 2023) | 11,758 | 519,000 | 10 | 21.8% | 40% | Work Collaboration |
| Questions (Platonov et al., 2023) | 48,921 | 153,540 | 301 | 3.0% | 52% | Question Answering |
| T-Finance (Tang et al., 2022) | 39,357 | 21,222,543 | 10 | 4.6% | 50% | Transaction Record |
| Elliptic (Weber et al., 2019) | 203,769 | 234,355 | 166 | 9.8% | 50% | Payment Flow |
| DGraph-Fin (Huang et al., 2022) | 3,700,550 | 4,300,999 | 17 | 1.3% | 70% | Loan Guarantor |
| T-Social (Tang et al., 2022) | 5,781,065 | 73,105,508 | 10 | 3.0% | 40% | Social Friendship |

*Table 6.* Statistics of datasets used for evaluation. Percentage of anomalies and the respective train splits have been mentioned.

### E.2. Hyperparameter Tuning

| Hyperparameter | Tuning Configurations | Description |
|---|---|---|
| $\mathcal{F}$ | $\{3, 5, 8, 10, 20, 25\}$ | Total number of graph filters. |
| $\delta$ | $\{0.2, 0.5, 0.7\}$ | Neighbourhood weight distribution parameter for `ORC` |
| $d_{\mathcal{P}}$ | $\{32, 48, 64, 128, 256\}$ | Total dimension of the product manifold. |
| `dropout` | $\{0.2, 0.3, 0.5\}$ | Dropout rate |
| `epochs` | $\{100, 150, 1000\}$ | Number of training epochs |
| `lr` | $\{1e-4, 2e-3, 0.001, 0.01\}$ | Learning rate |
| `weight_decay` | $\{0, 1e-4, 5e-4\}$ | Weight decay |

*Table 7.* Hyperparameter configurations used in the experiments for `CurvGAD`. We highlight the final configuration of `CurvGAD` in Red.

# F. Signature Estimation

The mixed-curvature product manifold $\mathbb{P}^{d_{\mathcal{P}}}$ is fundamental in capturing the geometric characteristics of graph-structured data. The optimal configuration of hyperbolic, spherical, and Euclidean components within this manifold is *dataset-dependent*, influenced by the underlying curvature distribution of the graph. To generalize across datasets, we estimate an appropriate manifold signature by leveraging Ollivier-Ricci Curvature (`ORC`) as a guiding metric.

Our heuristic-driven approach assigns a spherical component to datasets exhibiting a predominance of positively curved edges, while a hyperbolic component is prioritized for datasets with a surplus of negatively curved edges. The stepwise procedure for this signature selection is formalized in Algorithm 2 [`citation-withheld`][2]. By systematically analyzing the curvature distribution, our algorithm approximates a suitable manifold signature that aligns with the dataset's intrinsic geometric structure.

---

[2][`citation-withheld`] is used to respect the double blind policy for one of our previous works, which is still under review at the time of submission. Appropriate citation shall be inserted later. This algorithm has been adopted from our previous work.

To ensure robustness, we cluster the ORC distribution, identifying curvature centroids while preserving their sequence and frequency. This heuristic approach provides a structured methodology to inform curvature-aware representation learning. However, due to the inherent variability in optimal dimension allocation, we treat the dimensions of component manifolds as a tunable hyperparameter rather than an automatically inferred property. This strategy ensures fair and reproducible comparisons across datasets, as the ideal manifold configuration varies per dataset. While we do not claim that this heuristic yields the absolute optimal manifold decomposition, it serves as an effective approximation, aligning curvature-sensitive embeddings with the dataset's topological properties to enhance graph anomaly detection.

---

**Algorithm 2** Product manifold signature estimation

---

**Require:**  • Edge curvature histogram $\mathcal{C} = \{(\kappa_i, f_i)\}_{i=1}^N$
  • Threshold $\epsilon'$ to distinguish between curved and flat regions
  • Maximum number of Hyperbolic ($\mathcal{H}_{\max}$) and Spherical ($\mathcal{S}_{\max}$) components.
  • Total product manifold dimension $d_\mathcal{P}$
  • (Optional) Preferred component manifold dimensions $d_{(h)}^{\text{pre}}, d_{(s)}^{\text{pre}}, d_{(e)}^{\text{pre}}$

**Ensure:** Product manifold signature $\mathbb{P}^{d_\mathcal{P}} = \times_{p=1}^{\mathcal{P}} \mathcal{M}_q^{\kappa_p, d_p}$

1: Normalize frequencies: $f_i' = \frac{f_i}{\sum_{j=1}^N f_j}$
2: Construct weighted curvature set: $\mathcal{C}' = \{(\kappa_i, f_i')\}_{i=1}^N$
3: Determine optimal number of clusters $K$ using methods like the elbow method, constrained by $K \leq \mathcal{H}_{\max} + \mathcal{S}_{\max} + 1$ ▷ There can be only 1 Euclidean component
4: Cluster $\mathcal{C}'$ into $K$ clusters using weighted clustering (e.g., weighted K-means)
5: Initialize empty lists $\mathcal{H}, \mathcal{S}, \mathcal{E}$
6: **for** each cluster $c$ in clusters **do**
7:     Compute cluster centroid curvature $\kappa_c = \frac{\sum_{(\kappa_i, f_i') \in c} \kappa_i}{|c|}$
8:     Compute total frequency weight $w_c = \sum_{(\kappa_i, f_i') \in c} f_i'$
9:     **if** $\kappa_c < -\epsilon'$ **and** $|\mathcal{H}| \leq \mathcal{H}_{\max}$ **then**            ▷ Negative curvature
10:         Assign manifold component: $\mathcal{M}_p = \mathbb{H}^{\kappa_c}$            ▷ Curvature initialization
11:         Add $(\mathcal{M}_p, w_c)$ to $\mathcal{H}$
12:     **else if** $\kappa_c > \epsilon'$ **and** $|\mathcal{S}| \leq \mathcal{S}_{\max}$ **then**            ▷ Positive curvature
13:         Assign manifold component: $\mathcal{M}_p = \mathbb{S}^{\kappa_c}$            ▷ Curvature initialization
14:         Add $(\mathcal{M}_p, w_c)$ to $\mathcal{S}$
15:     **else**
16:         Assign manifold component: $\mathcal{M}_p = \mathbb{E}$            ▷ Approximate zero curvature, i.e. $\kappa_c \in [-\epsilon', \epsilon']$
17:         Add $(\mathcal{M}_p, w_c)$ to $\mathcal{E}$
18:     **end if**
19: **end for**
20: **if** Predefined dimensions $d_{(h)}^{\text{pre}}, d_{(s)}^{\text{pre}}, d_{(e)}^{\text{pre}}$ are provided **then**
21:     Assign dimensions $d_p$ to each component $p$ as per predefined values            ▷ Dimension assignment
22: **else**
23:     Set total number of components $\mathcal{Q} = |\mathcal{H}| + |\mathcal{S}| + |\mathcal{E}|$            ▷ Dimension assignment
24:     Allocate dimensions $d_p$ to each component $p$: $d_p = \left\lfloor d_\mathcal{P} \times \frac{w_p}{\sum_{p=1}^{\mathcal{Q}} w_p} \right\rfloor$            ▷ Proportional to weights
25:     Adjust $d_p$ to ensure $\sum_{q=1}^{\mathcal{P}} d_p = d_\mathcal{P}$
26: **end if**
27: Formulate manifold signature:
$$\mathbb{P}^{d_\mathcal{P}} = \left(\times_{h=1}^{|\mathcal{H}|} \mathbb{H}_h^{\kappa_{(h)}, d_{(h)}}\right) \times \left(\times_{s=1}^{|\mathcal{S}|} \mathbb{S}_s^{\kappa_{(s)}, d_{(s)}}\right) \times \mathbb{E}^{d_{(e)}}$$

---

