# OpenReview forum: "CurvGAD: Leveraging Curvature for Enhanced Graph Anomaly Detection"
_ICML.cc/2025/Conference — ICML 2025 poster_

### Official Review · Reviewer_24eB · 2025-03-14

**Overall Recommendation:** 3

**Summary:**

The paper introduces CurvGAD, a novel graph anomaly detection method that leverages mixed-curvature geometry to detect anomalies overlooked by conventional structural and attribute-based approaches. CurvGAD employs two parallel reconstruction pipelines: a curvature-equivariant pipeline that captures geometric anomalies using mixed-curvature embeddings, and a curvature-invariant pipeline utilizing Ricci flow for structural and attribute anomalies. The approach demonstrates superior anomaly detection, achieving up to 6.5% improvement in AUROC scores over state-of-the-art methods across ten diverse datasets.

**Claims And Evidence:**

The claims regarding the superiority of CurvGAD in detecting anomalies are well-supported by extensive experimentation and comparative analyses (Table 1). However, the claim of interpretability through curvature-based anomaly detection is partially supported only qualitatively (Figure 1); a quantitative interpretability evaluation is absent, weakening this claim.

**Essential References Not Discussed:**

The paper extensively cites the relevant literature, including critical references in mixed-curvature graph embeddings and anomaly detection. However, discussion on recent works involving adversarial robustness in geometric and spectral graph methods, such as those addressing structural robustness or attacks on anomaly detection systems, would strengthen the context.

**Experimental Designs Or Analyses:**

The experimental design, including the methodology for dataset splits, anomaly scoring, and evaluation metrics (Sections 5.3 and 5.4), is sound. One minor weakness is the absence of a detailed discussion regarding the computational overhead introduced by Ricci flow regularization across large datasets. More transparency regarding runtime comparisons with baselines would enhance confidence in practical applicability.

**Methods And Evaluation Criteria:**

The proposed methods, including curvature-equivariant and curvature-invariant reconstructions, effectively address the stated limitations. The evaluation using AUROC across ten datasets (Table 1) is appropriate and thorough, adequately capturing the method's efficacy in various contexts. The chosen baseline comparisons are relevant and robust, allowing for a clear assessment of CurvGAD’s relative performance.

**Other Comments Or Suggestions:**

The writing is clear, though a few typos are present (e.g., "compliment" instead of "complement" in Section 5.5 (e)). Also, improving readability in Figures 1 and 3 through clearer legends or annotations would benefit interpretation.

**Other Strengths And Weaknesses:**

The paper exhibits clear originality and significance in utilizing curvature for anomaly detection, offering substantial practical implications and interpretability. However, a major weakness is the lack of detailed hyperparameter settings for baseline methods, which limits reproducibility.

**Questions For Authors:**

How does computational complexity scale with large graphs, and is CurvGAD scalable enough for real-world networks containing millions of nodes?

Have you considered robustness analyses against adversarial perturbations, given the relevance to anomaly detection tasks?

**Relation To Broader Scientific Literature:**

CurvGAD is well-positioned within the context of existing literature on graph anomaly detection, clearly distinguishing itself by incorporating mixed-curvature and geometric perspectives (Section 2). While the paper cites recent works effectively, an expanded discussion on how CurvGAD’s geometric perspective fundamentally differs or extends beyond existing curvature-based embedding methods could further clarify its novelty

**Theoretical Claims:**

The paper does not present new theoretical proofs, but it effectively builds upon existing theoretical foundations of Riemannian geometry and Ricci curvature. The correctness and definitions in sections 3 and 4, particularly related to curvature and Ricci flow, appear mathematically sound and well-explained.

---

> ### Author Rebuttal · Authors · 2025-04-01
>
> We sincerely thank the reviewer for their insightful feedback, valuable assessment and constructive suggestions.
>
> ### **[Q1] Scalability and Complexity**
>
> Thank you for this important point. We provide a detailed time complexity (and scalability) analysis in **Appendix C**. Specifically:
> 1.  **Appendix C.1.2** explains our linear-time approximation of Ollivier-Ricci Curvature (ORC), which scales as $\mathcal{O}(|\mathcal{E}|)$ and is fully parallelizable.
>
> **Summary of Preprocessing Complexity (Appendix C.1.4)**
> - Ollivier-Ricci Curvature (ORC): $\mathcal{O}(|\mathcal{E}|)$ (linear time)
> - Ricci Flow Regularization: $\mathcal{O}(|\mathcal{E}|)$ (empirically bounded by a constant number of iterations)
> - Total Preprocessing Complexity: $\mathcal{O}(|\mathcal{E}|)$ (**parallelizable** and scalable)
>
>
> 2. **Appendix C.2** presents runtime comparisons (Table 5, Figure 4), showing CurvGAD’s efficiency across diverse datasets. Figure 4 lays down the average runtime per epoch comparision of CurvGAD with other baselines.
>
>
> The linear time approximation of ORC ensures that our model is scalable to million-scale graphs. Despite using multiple curvature manifolds and filters, CurvGAD achieves runtime comparable to or better than baselines like BGNN, XGBOD, and KGCN. For example, on Reddit, CurvGAD is faster than KGCN and significantly more efficient than XGBOD, while being far more expressive. Thus, our design ensures both scalability and geometric fidelity for large graphs.
>
>
> ### **[Q2] Robustness to adversarial perturbations?**
>
> We appreciate this insightful suggestion. Robustness to adversarial attacks is a valuable direction—especially for anomaly detection—but it lies beyond the current scope of this work. CurvGAD focuses on the role of mixed-curvature geometry in identifying anomalies. Our results establish curvature as a fundamental tool for GAD and pave the way for future advancements in geometry-aware graph learning while of- fering new avenues for exploring geometric anomalies in topologically complex networks. We are actively exploring robustness extensions in future work.
>
> ### **Other suggestions**
>
> - **Typos & Visuals**: Thank you for pointing these out. We will fix the typo (“compliment” → “complement” in Section 5.5(e)) and enhance Figures 1 & 3 with clearer legends and annotations in the camera-ready version.
> - **Baseline Hyperparameters**: We acknowledge this concern and will include detailed hyperparameter settings for all baselines in the camera-ready version and code repository. Due to the rebuttal length constraint, these could not be included here.

---

### Official Review · Reviewer_PTSu · 2025-03-14

**Overall Recommendation:** 2

**Summary:**

The paper proposes CurvGAD, a novel graph anomaly detection framework that incorporates curvature information through a mixed-curvature graph autoencoder consisting of two parallel pipelines: curvature-equivariant geometry reconstruction and curvature-invariant structure and attribute reconstruction.

**Claims And Evidence:**

yes

**Essential References Not Discussed:**

no

**Experimental Designs Or Analyses:**

yes

**Methods And Evaluation Criteria:**

yes

**Other Comments Or Suggestions:**

1. What is the special property of curvature over other metrics on graphs that is required for anomaly detection. In other words, there are lots of metrics that can capture geometric information, why do you choose curvature instead of others for anomaly detection, what makes it better than others?

2. "...under the homophily assumption (i.e., low-pass filters) and fail in heterophilic graphs where..." Homophily and low-pass filter are two different concepts. I think you've mixed them up. See [1] for the definition and explanation.

3. Is there anything new about equation (1)? Why do you use "mobius addition, mobius subtraction, κ-right-matrix-multiplication and κ-left-matrix multiplication" in equation (2-4), what is your motivation and what is their relations with graph anomaly detection? Are these your novel contributions or you borrow from other papers?

4. "filter bank captures both high-frequency (heterophilic) and low-frequency (homophilic) signals" It is not true that high-frequency signals can capture heterophilic information and low-frequency signals capture homophilic information. See the analysis and results in [2]

5. "enabling CurvGAD to generalize effectively across heterogeneous and homogeneous graphs." Heterogeneous/homogeneous or heterophilic/homophilic? See their differences in [3].

6. The heterophilic graphs used in experiments are not good enough as shown in [1]. Tolokers and Questions are actually homophilic graphs and both lead to beneficial effects to message passing instead of negative effects. Therefore, some analyses and conclusions in this paper need to be carefully re-examined based on correct evaluation and evidence. You need to use the real challenging heterophilic graphs, especially the malignant and ambiguous heterophily datasets identified in [1].




[1] The heterophilic graph learning handbook: Benchmarks, models, theoretical analysis, applications and challenges. arXiv preprint arXiv:2407.09618. 2024 Jul 12.

[2] When do graph neural networks help with node classification? investigating the homophily principle on node distinguishability. Advances in Neural Information Processing Systems. 2024 Feb 13;36.

[3] Beyond homophily in graph neural networks: Current limitations and effective designs. Advances in neural information processing systems. 2020;33:7793-804.

**Other Strengths And Weaknesses:**

see below

**Questions For Authors:**

see above

**Relation To Broader Scientific Literature:**

graph anomaly detection

**Theoretical Claims:**

no

---

> ### Author Rebuttal · Authors · 2025-04-01
>
> We sincerely thank the reviewer for their detailed and thoughtful comments. We deeply value the opportunity to clarify our contributions and improve our work based on your feedback.
>
> ### **[Q1] Why curvature?**
>
> The core aim of our work is to learn node representations on a **non-Euclidean manifold** and study how the underlying geometry—characterized by **curvature**—reveals anomalies.  While several geometric metrics exist, curvature helps in quantifying the intrinsic structural irregularities in graphs (our hypothesis for this paper).
>
> Unlike static scalar descriptors (e.g., clustering coefficients or local assortativity), curvature allows us to reason about how node-pair relations behave in a global geometric context, making it suitable for tasks like anomaly detection that are sensitive to subtle perturbations in local structure. Furthermore, graph regularization via Ricci Flow allows us to isolate non-geometric anomalies for reconstruction. If there are specific alternative geometric properties you had in mind, we would be eager to understand and consider them in future work.
>
> ### **[Q2, Q4] Homophiliy (Heterophily) and Low-pass (High-pass) Filtering**
>
> We appreciate this clarification and fully agree that homophily and low-pass filtering are not the same concept. We apologize if this was inadvertently conflated in the phrasing. Our intention was not to equate them, but to note that low-pass filters are particularly effective in capturing homophilic signals, which tend to lie in the low-frequency **eigenspectrum** of the graph Laplacian (and high-pass for heterophilic).
>
> To support our statement, we sincerely refer to the source cited by the reviewer [1], which states:
>
> - *“High-frequency signal, which can catch the neighborhood dissimilarity, is found to be helpful for heterophily.”*
> - *“It is commonly stated that low-pass filter mainly captures the smooth components of the signals (neighborhood similarity), while high-pass filter mainly extracts the non-smooth ones (neighborhood dissimilarity).”*
>
> Thus, we only argue that low-pass filters assist with homophilic patterns and high-pass filters help capture heterophilic patterns, which aligns with the aforementioned literature. We will revise our phrasing in the CR version to avoid ambiguity.
>
> [1] The heterophilic graph learning handbook. 2024 Jul 12.
>
> ### **[Q3] Novelty of Equations 2-4**
>
> Yes, Equations (2–5) are indeed novel contributions of our work. They extend Chebyshev spectral filtering to the mixed-curvature Riemannian setting, and define the filterbank using the $\kappa$-stereographic model. Specifically:
> - Eq. (2): Projects Euclidean features to the manifold.
> - Eq. (3–4): Define manifold-aware Chebyshev filtering and aggregation. They extend the typical Chebyshev recurrence to the mixed-curvature spaces.
>
> These gyrovector operations (Mobius addition, $\kappa-$matrix multiplication) are standard in non-Euclidean geometry and are described in Appendix B.2. They help us extend the Euclidean operations to the product manifold.
>
> ### **[Q5] Heterogeneous or heterophilic?**
>
> This is correct and we apologize for the typo (and therefore, the unintended abuse of terminology). We indeed meant "heterophilic" and "homophilic", not "homogeneous" or "heterogeneous" in the sentence mentioned. We will rectify in the CR version.
>
> ### **[Q6] Additional Experimentation on Heterophilic Networks**
>
> Thank you for this valuable suggestion. We initially evaluated CurvGAD on a diverse mix of homophilic and heterophilic benchmarks:
> - Homophilic: Reddit, Weibo, Amazon, YelpChi, Tolokers, Questions
> - Heterophilic: T-Finance, Elliptic, DGraph, T-Social
>
> Based on the feedback we conducted additional experiments on heterophilic datasets from [1]:
> - Malignant: Cornell, Texas
> - Ambiguous: snap-patents, twitch-gamers
>
> As these datasets lack organic anomaly labels, we follow a common setup by treating the minority class as anomalous. We compare CurvGAD against strong (select) baselines (due to spatial and time constraints): GCN, ADAGAD, KGCN, and BWGNN.
>
> #### **AUROC (%) Results on more Heterophilic Datasets**
>
> | Dataset             | GCN                | ADAGAD         | KGCN           | BWGNN          | **CurvGAD (Ours)** |
> |-------------------  |----------------|----------------|----------------|----------------|---------------------|
> | **Cornell**        | 72.3 ± 1.8       | 73.5 ± 2.1     | 78.8 ± 1.9     | 75.2 ± 2.9     | **84.6 ± 1.7**      |
> | **Texas**          | 74.5 ± 0.3        | 78.2 ± 0.2     | 79.9 ± 2.0     | 81.3 ± 1.2     | **85.7 ± 0.6**      |
> | **snap-patents**   | 41.2 ± 1.1     | 49.5 ± 1.5     | 48.5 ± 0.4     | 49.1 ± 1.2     | **52.03 ± 0.9**      |
> | **twitch-gamers**  | 66.8 ± 1.9        | 71.1 ± 2.4     | 73.3 ± 2.1     | 75.2 ± 2.1     | **78.34 ± 1.1**      |
>
> These results demonstrate CurvGAD’s generalization to heterophilic graphs. We will include full analysis (all baselines) in the CR version.

---

### Official Review · Reviewer_BDH1 · 2025-03-15

**Overall Recommendation:** 4

**Summary:**

The authors propose a method for graph anomaly detection that directly takes into account the curvature of the graph, thus capturing the structure of the graph in an arguably more appropriate way than what can be done with Euclidean representations (i.e., the common approach of embedding nodes/graphs via GNNs). The proposed approach, CurvGAD uses 1) a mixed-curvature encoder that captures irregularities in curvature and 2) an encoder-decoder module based on Ricci Flow to capture the structure of the adjacency matrix and the node features.

Experimental results show a substantial improvement in graph anomaly detection (posed as a classification task) compared to several strong competitors.

## update after rebuttal
After going through the other reviews and the response from the authors, I will maintain my original score. I was initially (and still am) overall positive on this paper, and I have not found any major counterpoints in the other reviews.

**Claims And Evidence:**

Yes.

**Essential References Not Discussed:**

No

**Experimental Designs Or Analyses:**

Yes. No issues.

**Methods And Evaluation Criteria:**

Yes.

**Other Comments Or Suggestions:**

Call me a killjoy, but I don't think "my model wins" is appropriate in a scientific paper. I would suggest choosing a different caption for Table 1.

**Other Strengths And Weaknesses:**

Weaknesses:
- Computing ORC is computationally expensive

**Questions For Authors:**

NA

**Relation To Broader Scientific Literature:**

This paper contributes to the graph anomaly detection literature. This is a very active area of research. However, methods that consider the curvature are still not very commonly used.

**Theoretical Claims:**

Not much to check. The model leverages existing theory.

---

> ### Author Rebuttal · Authors · 2025-04-01
>
> We sincerely thank the reviewer for their insightful feedback and constructive suggestions. We hereby address the raised concerns.
>
> ### **Computational Considerations for ORC**
>
> > Computing ORC is computationally expensive
>
> We appreciate your concern and apologize for not sufficiently emphasizing this aspect in the main paper. To clarify, we employ a **linear-time combinatorial approximation** for computing Ollivier-Ricci Curvature (ORC), which avoids the expensive optimal transport computations typically involved. A detailed analysis of the computational considerations for ORC is provided in **Appendix C.1**, particularly in **Section C.1.2**.
>
> As outlined in **Theorem C.1** and Appendix C.1, our approximation computes curvature using only local structural properties—specifically node degrees and triangle counts—allowing each edge's ORC to be estimated as the average of analytically derived upper and lower bounds. This approach results in a **linear-time** computational complexity $(\mathcal{O}(|\mathcal{E}|))$. Furthermore, because the computation for each edge is independent, the process is naturally parallelizable, making it scalable and efficient even for large graphs.
>
> **Summary of Preprocessing Complexity (Appendix C.1.4)**
> - Ollivier-Ricci Curvature (ORC): $\mathcal{O}(|\mathcal{E}|)$ (linear time)
> - Ricci Flow Regularization: $\mathcal{O}(|\mathcal{E}|)$ (empirically bounded by a constant number of iterations)
> - Total Preprocessing Complexity: $\mathcal{O}(|\mathcal{E}|)$ (**parallelizable** and scalable)
>
>
> > I don't think "my model wins" is appropriate in a scientific paper.
>
> We thank the reviewer for pointing this out. The caption could be more formal and we will revise it to adopt a more appropriate and professional tone in the camera-ready version.

---

> > ### Comment · Reviewer_BDH1 · 2025-04-04
> >
> > I thank the authors for addressing my comments. After reading their response and the other reviews, I will maintain my original score.

---

### Official Review · Reviewer_Vzbi · 2025-03-15

**Overall Recommendation:** 3

**Summary:**

This paper proposed a mixed-curvature graph autoencoder that detects curvature-based geometric anomalies. It combines geometry reconstruction, using a Riemannian encoder and Gaussian kernel-based decoder, with structure and attribute reconstruction, regularizing graph curvature through discrete Ollivier-Ricci flow to isolate non-geometric anomalies. This approach refines anomaly classifications and uncovers new curvature-driven anomalies.

**Claims And Evidence:**

The authors claim that curvature plays a crucial role in graph anomaly detection, supported by evidence in Figure 1. The motivations outline three key challenges, each backed by citations to the corresponding supporting evidence.

**Essential References Not Discussed:**

The references are comprehensive and well-discussed.

**Experimental Designs Or Analyses:**

The experimental settings, including datasets and evaluation criteria, align with existing works.

**Methods And Evaluation Criteria:**

The experimental settings, including datasets and evaluation criteria, make sense.

**Other Comments Or Suggestions:**

No more suggestions. The paper is well-written.

**Other Strengths And Weaknesses:**

**Strength:**

1)The paper is well-written, and well-organized.

2)The experimental results are comprehensive.

3)The motivation is justified and clear.

**Weaknesses:**

I still have some questions:

1)The authors claim that "the input graph needs to be regularized to standardize the curvature of the graph, which allows the subsequent structure and attribute reconstructions to focus solely on non-geometric anomalies." Could the authors provide visualizations or quantitative results showing the curvature of the regularized graph to support that this operation effectively removes geometric information?

2)The authors state that CurvGAD can handle both heterophilic and homophilic networks, with the primary contribution focusing on addressing heterophilic problems. How does the proposed model perform on homophilic anomalies? Which specific components of the model contribute to its effectiveness in such cases?

3)Does this curvature difference extend to non-graph data? If so, could the approach be generalized to other types of data structures?

4)The proposed method includes two scores for anomaly detection. Are these scores normalized to the same scale? What happens if the scores differ significantly, potentially causing one of them to fail? Have the authors considered such a scenario, and if so, how is it addressed?

**Questions For Authors:**

Please refer to Weaknesses.

**Relation To Broader Scientific Literature:**

The contributions and motivations  related references are all listed in this manuscript intuitively.

**Theoretical Claims:**

Not related.

---

> ### Author Rebuttal · Authors · 2025-04-01
>
> We sincerely thank the reviewer for their positive assessment and thoughtful questions. Please find our detailed responses below:
>
> > [Q1] Could the authors provide visualizations or quantitative results showing the curvature of the regularized graph?
>
> We thank the reviewer for pointing this out. We would like to clarify that **Figure 3** in the main paper visualizes the Karate Club graph before and after Ricci Flow regularization. The edge curvatures are color-coded—red/blue indicating positive/negative curvature before the flow, and grey edges (≈0 curvature) after the flow. This demonstrates that Ricci Flow effectively neutralizes geometric irregularities, enabling subsequent modules to focus on non-geometric anomalies. We shall add similar visualisations (and their corresponding curvature values) for other example graphs in the camera ready version.
>
> > [Q2]  Which specific components of the model contribute to its effectiveness in such cases? (Homophily)
>
> We appreciate this insightful question. CurvGAD is equipped to handle both homophilic and heterophilic settings through a learned spectral **filter bank**, as described in **Section 4.1.1**. This bank consists of:
> - Low-pass filters to capture smooth, homophilic signals,
> - High-pass filters for heterophilic patterns, and
> - An identity (unfiltered) pathway to retain raw structural information.
>
> These filters are assigned learnable weights, enabling the model to adaptively emphasize the appropriate frequency bands depending on the graph type. As a result, CurvGAD is able to generalize across both homophilic and heterophilic graphs by **selectively attending** to different parts of the **eigenspectrum**, which is reflected in its strong performance on diverse benchmarks.
>
> > [Q3] Does this curvature difference extend to non-graph data?
>
> The idea of using curvature as a measure of structural irregularity indeed extends beyond graphs, particularly in data types that can be embedded into metric or manifold spaces, such as text, images, or time series. While such extensions are beyond the scope of the current paper, we are actively exploring them in future work.
>
> > [Q4] The proposed method includes two scores for anomaly detection.
>
> We appreciate the opportunity to clarify this. Yes, all the three reconstruction-based anomaly scores (curvature, features, adjacency matrix) are normalised using min-max scaling. We shall clarify and highlight this more explicitly in the camera-ready draft and code repository. We also use learnable trade-off weights $(\lambda_{\mathbf{C}}, \lambda_{\mathbf{A}}, \lambda_{\mathbf{X}}, \lambda_{\text{cls}})$ to adaptively balance the contributions of all losses. This ensures that no single score dominates (**Section 4.3**). We further perform extensive ablation studies to isolate the impact of different scores (Table 2). Section 5.5 entails the different CurvGAD variants which *omit specific losses* out of these four scores.
>
> We sincerely thank the reviewer again for their thoughtful and constructive feedback, which will help us further improve the clarity and completeness of our work.

---

### Decision · Program_Chairs · 2025-05-01

**Decision:**

Accept (poster)

**Comment:**

Four reviewers have evaluated this paper and scored it 2, 3, 3, 4. Overall, the reviewers had a number of concerns ranging from the issues with computational efficiency (some remained unconvinced), whether the method is purely applicable to graphs only and issues with the abuse of terminology (e.g., confusion between homophily and low-pass filtering). On balance, while the paper is borderline, the rebuttal is mostly sound and the AC recommends a weak accept.